# An investigation of the global uptake of CO₂ by lime from 1930 to 2020

Longfei Bing[1,2,3]★, Mingjing Ma[1,4]★, Lili Liu[5], Jiaoyue Wang[1,2,3], Le Niu[1,4] and Fengming Xi[1,2,3]

[1]Institute of Applied Ecology, Chinese Academy of Sciences, Shenyang 110016, China

[2]Key Laboratory of Pollution Ecology and Environmental Engineering, Chinese Academy of Sciences, Shenyang 110016, China

[3]Key Laboratory of Terrestrial Ecosystem Carbon Neutrality, Liaoning Province, Shenyang 110016, China

[4]University of Chinese Academy of Sciences, Beijing 100049, China

[5]Search CO₂ (Shanghai) Environmental Science & Technology Co., Ltd, Shanghai 200232, China

★These authors contributed equally to this work.

*Correspondence*: Fengming Xi (xifengming@iae.ac.cn)

**Abstract.** A substantial amount of $CO_2$ is released into the atmosphere from the process of the high temperature decomposition of limestone to produce lime. However, during the life cycle of lime production, the alkaline components of lime will continuously absorb $CO_2$ from the atmosphere during use and waste disposal. Here, we adopt an analytical model describing the carbonation process to obtain regional and global estimates of carbon uptake from 1930 to 2020 using lime lifecycle use-based material data. The results reveal that the global uptake of $CO_2$ by lime increased from 9.16 Mt C yr⁻¹ (95% confidence interval, CI: 1.84-18.76 Mt C) in 1930 to 35.27 Mt C yr⁻¹ (95% CI: 23.63-50.39 Mt C) in 2020. Cumulatively, approximately 1444.70 Mt C (95% CI:1016.24-1961.05 Mt C) were sequestered by lime produced between 1930 and 2020, corresponding to 38.83% of the process emissions during the same period, mainly contributed from the utilisation stage (76.21% of the total uptake). We also fitted the missing lime output data of China from 1930 to 2001, thus compensating for the lack of China's lime production (cumulative 7023.30 Mt) and underestimation of its carbon uptake (467.85 Mt C) in the international data. Since 1930, lime-based materials in China have accounted for the largest proportion (about 63.95%) of the global total. Our results provide data to support including lime carbon uptake into global carbon budgets and scientific proof for further research of the potential of lime-containing materials in carbon capture and storage. The data utilized in the present study can be accessed at https://doi.org/10.5281/zenodo.7759053 (Ma, 2023).

## 1 Introduction

According to the latest report (6th Assessment) of the Intergovernmental Panel on Climate Change (IPCC), anthropogenic activities are responsible for the unprecedented increase in the concentration of $CO_2$ in the atmosphere, which reached 415 ppm in 2021 (NOAA. ESRL, 2022.). In 2019, approximately 24% (14 Gt CO₂-eq) of the net global anthropogenic emissions

originated from industrial sources, and lime production was the second highest industrial source after cement production (IPCC,

2021; Shan et al., 2016). Similar to cement, lime is mainly produced via the heating of limestone ($CaCO_3$) in a kiln at temperatures of 900–1200 °C. The $CO_2$ generated during this process is commonly released into the atmosphere (Greco-Coppi et al., 2021). During limestone decomposition, fossil fuel combustion, which is used to provide energy for the process, is an indirect source of $CO_2$, but this is often accounted for in the energy sector (IPCC, 2021).

The enormous quantity of lime produced in the world (~427 Mt in 2020; USGS, 2022) is mainly employed in the following

sectors (Figure 1): (1) chemical industry, such as for the production of precipitated calcium carbonate (PCC), manufacturing of paper, and refining of sugar; (2) environmental remediation/treatment, including water treatment, acid mine drainage, and flue gas desulphurization; (3) metallurgical industry, for instance as a fluxing agent in the production of iron and steel; and (4) construction industry for building materials including lime mortar and lime-stabilised soil-asphalt mixtures (National Lime Association, 2020). Many lime-based materials, including wastes produced in different industries, re-absorb some of the

released $CO_2$, and thereby sequester $CO_2$ throughout the lime cycle (carbonation), owing to the unstable calcium oxide in these materials (Cizer et al., 2012a). According to Renforth (2019), approximately 34% of lime can directly or indirectly remove $CO_2$ from the atmosphere and absorb $CO_2$ during the utilization stage. The carbonation process can be described using the following reactions:

$$CaO+H_2O=Ca(OH)_2 \tag{1}$$

$$Ca(OH)_2+CO_2=CaCO_3+H_2O \tag{2}$$

Carbonation proceeds progressively from the exterior to the interior of lime-containing materials via the diffusion of $CO_2$ into

particles, followed by its reaction with hydration products of calcium oxide (Cizer et al., 2012b; Despotou et al., 2016). Therefore, carbonation can be considered as a mineralisation technology for carbon capture, utilization, and storage (CCUS) (Lai et al., 2021; Snæbjörnsdóttir et al., 2020). Samari et al. (2020) indicated that lime-based materials have been proposed as solid sorbents in direct air capture (DAC) technologies (extraction $CO_2$ directly from the atmosphere). In practice, however, because of material and environmental factors, only 70–80% of the CaO in lime can be converted into $CaCO_3$ (Bhatia and

Perlmutter, 1983). In previous studies, the carbonation process and factors influencing its rate (Ma et al., 2019), as well as strategies for improving the sequestration of carbon using lime-containing materials under controlled laboratory conditions (Pan et al., 2012; Baciocchi, 2017), have been examined. Pan et al. (2020), for instance, estimated the $CO_2$ reduction potential of lime-based solid wastes (e.g., lime mud, red mud and iron and steel slags) in mineralisation technologies, and highlighted a substantial potential for the storage of $CO_2$ in these wastes. The maximum achievable carbonation capacity of these solid

wastes via direct mineralisation is approximately 310 Mt of $CO_2$ per year. Renforth (2019) estimated the global potential of $CO_2$ uptake through carbonation of lime and related alkaline materials up to the year 2100 (approximately 2.9–8.5 Gt of $CO_2$ per year) and indicated that this process can substantially mitigate $CO_2$ emissions during manufacturing of the associated materials. However, existing studies are limited to estimation of the carbon reduction potential via accelerated carbonation instead of carbon sequestration throughout the lime cycle under realistic conditions.

In the present study, a carbon sequestration analytical model was utilized to evaluate the global uptake of $CO_2$ by lime-containing materials during the three stages (production, use and waste disposal process) of the lime cycle from 1930 to 2020. The aims were to highlight the magnitude of the lime carbon sink on a global scale and to estimate the net $CO_2$ emission associated with the production of lime. In addition, characteristics of the uptake of $CO_2$ by lime and the contribution to the carbon cycle were examined. The present study significantly improves the global carbon uptake model and provides theoretical support for the utilization of lime-containing materials in carbon capture and storage (CCS).

## 2 Data and Methods

### 2.1 Lime production, resources usage proportion and treatment

In this study, China and the United States (U.S.) were considered individually, while all other producers were grouped together as "rest of the world" (ROW). The global lime production data came from Lime Statistics and Information (USGS, 2022), but the data did not include the statistics of China's lime production between 1963 and 1984. In addition, the statistical value of China's lime production from 1985 to 2001 was underestimated compared with the actual value (Cao et al., 2019), which led that the statistical data of global lime production during 1963-2001 was significantly less than the actual production (Fig. 2b). The lime production data of China in this study were obtained from (China Construction Material Industry Yearbook, 2022). Considering that lime production data is available for the United States since 1930, which is much earlier than the recorded data for China and the ROW, we filled gaps in the data using fitting methods, thereby extending the time scale of the study to 1930.

First, we fitted China's lime production. The only source of China's lime production statistics is the "China Building Materials Industry Yearbook", which records the lime production data from 1996 to 2020, of which the data from 2015 to 2018 is missing; in addition, the statistical yearbook introduces the use of lime in various industries. From this, we know that the production of lime in construction, steel, calcium carbide, and alumina in the downstream sector of lime accounts for more than 90% of lime production. Therefore, we collected data on China's cement production (1930–2020), the completed area of housing in the whole society (1963–2020), steel production (1949–2020), calcium carbide production (1949–2020), and alumina production (1954–2020) and fitted them to the lime production data. Taking China's lime production as the dependent variable, the stepwise linear regression method was used to construct a regression model. Since the completed area data of houses in the whole society was only available from 1963, the model predicted lime production data from 1963 to 1995. Then, through the ARIMA (0,1,0) model, with external control variables including the steel production, calcium carbide production, and cement production, we fit the lime production in China from 1949 to 1962 (the steel and calcium carbide production data were only extended to 1949). Finally, we used the ARIMA (2,2,0) model without external control variables to fit the lime production in China from 1930 to 1948. From this, we obtained the fitted lime production data for China from 1930 to 2020 (Fig 2a). Fitted coefficients of regression models and ARIMA models are shown in SI-2 Data 4.

After obtaining the Chinese lime production data, we corrected the global lime production data from the USGS from 1930 to

2020 (Fig 2b). The ARIMA (1,0,0) model was then used to fit the global lime production from 1930 to 1962 with global alumina production, steel production and cement production as external control variables.

Relatedly, according to data that were obtained from the USGS, approximately 15%–42% of lime resources in the U.S. are utilized in the chemical industry (mainly for petroleum refining and glass and rubber products production), whereas 30%–51% are employed in metallurgy (primarily in the production of crude steel), 5%–14% are used in the construction industry (principally for the production of lime stabilised soil and lime motor), and approximately 8%–43% are applied in environmental protection and other fields. In the ROW, data on the usage of lime resources in different sectors including the industry were mostly obtained from publications (see the SI-2 Data 9 and SI-3 Data 1).

## 2.2 Estimation of emissions from processes

Regarding industrial processes, lime production is the second-highest source of carbon emissions after cement production, and thus, its contribution cannot be ignored (Shan et al., 2016). $CO_2$ emissions from lime production are mainly linked to the calcination stage, during which calcium oxide (CaO or quicklime) is formed from the decomposition of limestone by heat (Despotou et al., 2016). Lime comes from the decomposition of limestone in shaft or rotary kiln, and the carbon emission of this industrial process can be estimated from using the IPCC method (IPCC, 2006). Considering the availability of lime production data, Method 1(multiplication of the regional lime production by the $CO_2$ emission factor) from the IPCC Guidelines for National Greenhouse Gas Inventories was utilized to calculate $CO_2$ emissions from lime production processes in the present, and this can be expressed as follows:

$$CE_{l,i} = m_{l,i} \times EF_l \tag{1}$$

where $CE_{l,i}$ is the annual $CO_2$ emissions, $m_{l,i}$ represents the production of the lime industry, and $EF_l$ denotes the $CO_2$ emission factor associated with the lime production process. l refers to different types of lime use, including PCC, sugar making, lime-stabilized soil and lime mortar, and i refers to different years.

Emission factors for the lime industry processes were determined using the composition of raw materials and the production technology. In the present study, 0.77-, 0.683-, and 0.75-ton $CO_2$ per ton of lime produced were adopted as emission factors for the US, China, and ROW, respectively (IPCC, 2006). Emission factors for the U.S. and ROW were according to the IPCC guidelines, whereas that for China was from the National Development and Reform Commission of China (Guidelines for provincial greenhouse gas inventories, 2011).

## 2.3 Assessments of uptake during the lime cycle

Lime materials, which remove $CO_2$ from the atmosphere, belong to the following stages of the lime cycle: (1) production, (2) service and (3) waste disposal (Fig. 1). Therefore, the $CO_2$ uptake by lime ($C_{l,total}$) was calculated using the following formula:

$$C_{l,total} = C_{l,pro} + C_{l,ser} + C_{l,wd} \tag{2}$$

where $C_{l,pro}$, $C_{l,ser}$, and $C_{l,wd}$ are the uptake components during the production, service and waste disposal stages, respectively. The uptake of $CO_2$ in different stages of the lime cycle is examined subsequently.

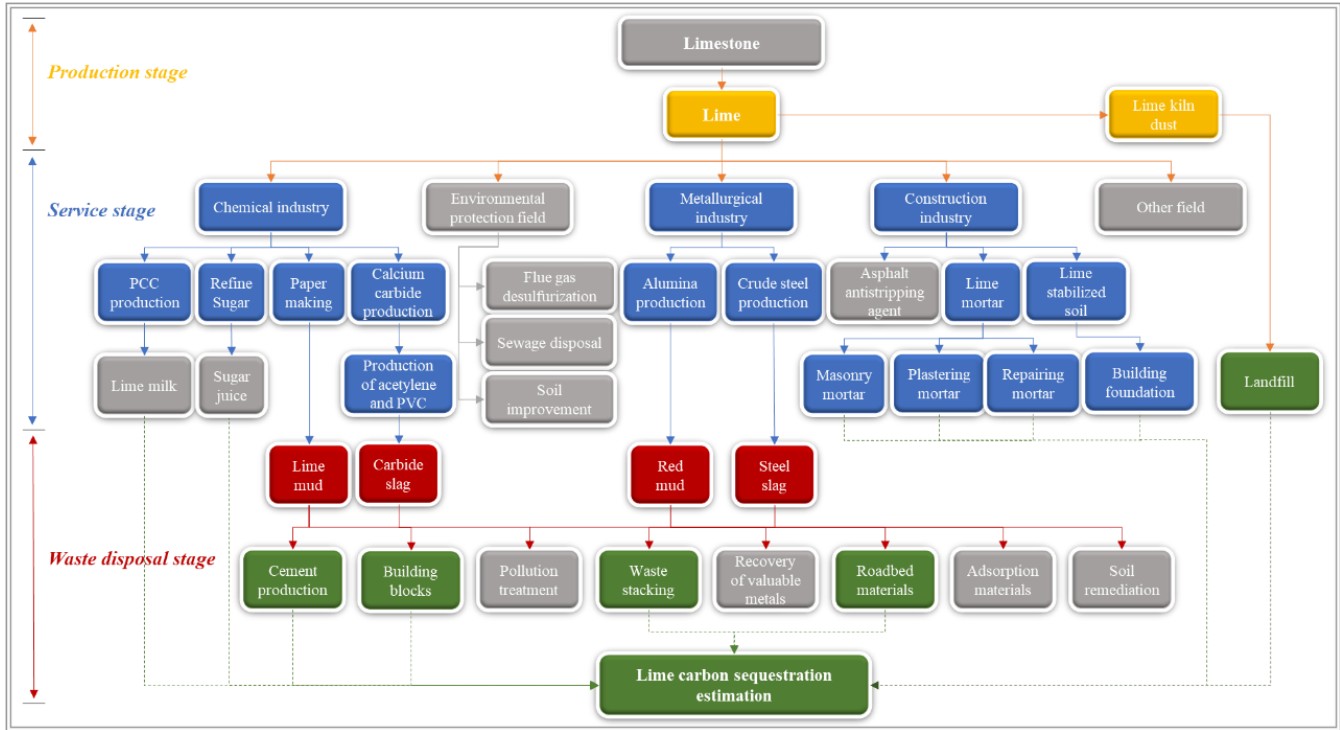

**Figure 1: System boundary for the sequestration of carbon by lime. Solid arrows represent the material flow, dashed arrows indicate the carbon flow. (Yellow, blue, and red represent lime-based materials with carbon absorption capacity and their associated production processes, spanning from initial production through usage and waste disposal. Gray represents materials, production processes, or disposal methods with little carbon absorption capacity. Green represents the disposal method for lime-based waste that possesses carbon absorption potential.)**

### 2.3.1 Assessment of uptake during the production stage

The carbon sink of the lime production stage refers to the uptake of $CO_2$ by lime kiln ash, and this can be quantified using the following expression:

$$C_{l,pro} = m_{l,i} \times r_{lkd} \times f_{lkd}^{CaO} \times \gamma_{lkd} \times \frac{M_C}{M_{CaO}} \tag{3}$$

where $m_{l,i}$ is the quantity of lime produced, $r_{lkd}$ represents the output rate of lime kiln ash, $f_{lkd}^{CaO}$ denotes the concentration of CaO in dust, $\gamma_{lkd}$ is the rate of conversion of CaO to $CaCO_3$ in dust, and $M_{CaO}$ and $M_{CO_2}$ are molar masses of CaO and C, which in the present study are 56 and 12 g/mol, respectively.

### 2.3.2 Assessment of uptake during the service stage

Processes that can absorb $CO_2$ in the lime utilization stage principally comprise the production of precipitated calcium carbonate (PCC, $C_{pcc,i}$), carbonation sugar (SUG, $C_{sug,i}$), lime-stabilised soil (LSS, $C_{lss,i}$), and lime mortar (MOR, $C_{mor,i}$). The uptake of $CO_2$ in this stage can be calculated as follows:

$$C_{l,ser} = C_{pcc,i} + C_{sug,i} + C_{lss,i} + C_{mor,i} \tag{4}$$

(1) PCC and SUG

PCC is produced via the hydration of high-calcium quicklime, followed by a reaction of the resulting slurry and $CO_2$ (Wang

*et al.*, 2002), and this reaction can be represented as follows: $Ca(OH)_2 + CO_2 = CaCO_3 \downarrow + H_2O$. According to the law of conservation of mass, the uptake of $CO_2$ by lime in PCC can be calculated as follows:

$$C_{pcc,i} = m_{l,i} \times L_1 \times a_1 \times f_l^{CaO} \times \frac{M_C}{M_{CaO}} \tag{5}$$

where $L_1$ is the proportion of lime that is used in the chemical industry, $a_1$ is the proportion of lime utilized in the chemical industry that is associated with PCC, and $f_l^{CaO}$ is the concentration of CaO in lime. Similar to the principle of the carbon sink in the production of PCC, the uptake of $CO_2$ linked to the production of carbonation sugars (SUG) can be calculated using the

145 following expression:

$$C_{sug,i} = m_{l,i} \times L_1 \times a_2 \times f_{sug} \times f_l^{CaO} \times \frac{M_C}{M_{CaO}} \tag{6}$$

where $a_2$ is the proportion of lime used in the production of SUG in the industry and $f_{sug}$ is the proportion of sugar produced using the lime-refining method.

(2) LSS

Under wet conditions, the carbonation rate of a LSS is approximately between 70%–80% over a duration of three months (Liu

et al., 2018b). Therefore, it is assumed that LSS can be carbonated within a year, and the uptake of $CO_2$ by LSS is quantified using the following expression:

$$C_{lss,i} = m_{l,i} \times L_2 \times a_3 \times f_l^{CaO} \times \gamma_{lss} \times \frac{M_C}{M_{CaO}} \tag{7}$$

where $L_2$ is the proportion of lime used in the construction sector, $a_3$ represents the proportion of lime employed in LSS in the construction sector, and $\gamma_{lss}$ is the rate of conversion of CaO to $CaCO_3$ in LSS.

(3) MOR

MOR is mostly used for the plastering of interior walls, with a typical thickness of 20 mm (Almanac of China building materials industry, 2022). Under natural conditions, the estimated carbonation rate of MOR is 1 mm d$^{-0.5}$ (Ventol et al., 2011) ). Therefore, according to Fick's law of diffusion, a year is insufficient for the complete carbonation of a MOR layer. Consequently, the uptake of $CO_2$ by MOR is calculated using the following expressions:

$$C_{mor,i} = m_{l,i} \times L_2 \times a_4 \times f_{mor,i} \times f_l^{CaO} \times \gamma_{mor} \times \frac{M_C}{M_{CaO}} \tag{8}$$

$$d_{mor}=k_{mor}\times\sqrt{t_{mor}} \tag{9}$$

$$f_{mor,i}=(d_{mor,i}-d_{mor,i-1})/d_T \tag{10}$$

where $L_2$ is the proportion of lime used in the construction sector, $a_4$ denotes the proportion of lime in MOR that is utilized in the construction sector, $f_{mor,i}$ represents the carbonation ratio of MOR in the i-th year, $\gamma_{mor}$ is the rate of conversion of CaO to $CaCO_3$ in MOR, $d_{mor,i}$ represents the depth of carbonation of MOR in the i-th year; $k_{mor}$ denotes the rate of carbonation of MOR, $t_{mor}$ is the duration of carbonation of MOR and $d_T$ is the thickness of MOR.

### 2.3.3 Assessment of uptake during the waste disposal stage

Lime employed in the production of paper, aluminium, calcium carbide, and steel generates by-products including lime mud (LM, $C_{LM,i}$), red mud (RM, $C_{RM,i}$), carbide slag (CS, $C_{cs,i}$), and steel slag (SS, $C_{ss,i}$), respectively. The alkaline component (CaO) in these wastes absorb $CO_2$ under natural conditions.

(1) LM and RM

Lime mud particles that are involved in the production of paper are usually fine and evenly distributed (Ma et al., 2021). In fact, particles < 40 μm account for 93%, and the associated water contents range from 39% to 60% (Qin et al., 2015). However, as a paste, the penetration of $CO_2$ to react with the lime mud is limited. Consequently, a year is usually insufficient for the complete carbonation of lime mud.

Red mud is also characterised by fine particles as well as a porous structure, high specific surface area, and good stability in water (Wang et al., 2019). Similar to the principle of the carbon sink for lime mud, a year is insufficient for the complete carbonation of red mud (Liu et al., 2018b). The uptake of $CO_2$ by lime in lime and red muds is calculated using the following expression:

$$\varepsilon_{m,ij}=m_{p,ij}\times r_{m,ij}\times f_{m,j}^{CaO} \tag{11}$$

where $\varepsilon_{m,ij}$ denotes the mass of CaO in wastes (m,j=lime mud or red mud) that can be carbonated in year i, $m_{p,ij}$ is the quantity of paper and paperboard/alumina that are produced in the i-th year, and p is the production. $r_{m,ij}$ is the output rate of waste j and $f_{m,j}^{CaO}$ represents the concentration of CaO in waste j.

According to Fick's law of diffusion, the depth of carbonation of waste j ($d_{m,ij}$) can be obtained from the carbonation rate ($k_{m,j}$) and carbonation time ($t_i$) using the following equations:

$$d_{m,ij}=k_{m,j}\times(\sqrt{t_i}-\sqrt{t_{i-1}}) \tag{12}$$

$$R_{m,ij}=\begin{cases}\dfrac{\dfrac{k_{m,j}\times\sqrt{t_i}}{h_{m,j}}}{t_{m,j}}\times t_i & (t_i\leq t_{m,j})\\[2ex] \dfrac{d_{m,ij}}{h_{m,j}} & (t_{m,j}<t_i<100)\end{cases} \tag{13}$$

where $R_{m,ij}$ represents the fraction of waste j that is carbonated in the i-th year, $h_{m,j}$ is the height of the waste j pile and $t_{m,j}$ is the duration of the yard of the waste j. Accordingly,

$$C_{m,ij}=\varepsilon_{m,ij} \times (1 - f_{m,ij}^{use}) \times R_{m,ij} \times \gamma_{m,j} \times \frac{M_C}{M_{CaO}} \tag{14}$$

where $C_{m,ij}$ is the uptake of $CO_2$ uptake of waste j during the i-th year, $f_{m,ij}^{use}$ denotes the utilization rate of waste j and $\gamma_{m,j}$ is the rate of conversion of CaO to $CaCO_3$ in lime mud.

(2) CS

Carbide slag comprises particles that are dominantly between 10–50 μm, which usually contain moderate amounts of water (Lin et al., 2006). Stacking for approximately 15 d can reduce the concentration of CaO by approximately 50% (Hao et al., 2013). The uptake of $CO_2$ by CS can be calculated using the following expressions:

$$\varepsilon_{cs,i}=m_{l,i} \times L_1 \times a_5 \times p_l^{cs} \times r_{cs} \times f_{cs}^{CaO} \tag{15}$$

$$C_{cs,i} = \varepsilon_{cs,i} \times (1 - f_{cs}^{use}) \times \gamma_{cs} \times \frac{M_C}{M_{CaO}} \tag{16}$$

where $\varepsilon_{cs,i}$ is the mass of CaO in CS in the i-th year, $a_5$ denotes the proportion of lime in calcium carbide that is utilized in
the chemical industry, $p_l^{cs}$ represents the output of calcium carbide per ton of lime input, $r_{cs}$ is the output rate of CS, $f_{cs}^{CaO}$ is the concentration of CaO in CS, $f_{cs}^{use}$ is the utilization rate of CS and $\gamma_{cs}$ is the rate of conversion CaO to $CaCO_3$ in CS.

(3) SS

SS cannot be carbonated within a year because its hydration commonly requires more than 4 years (Wang and Yan, 2010). In the present study, the SS particle was approximated as a uniformly-densified sphere. The fraction ($R_{s,i}$) of SS that is carbonated
can be estimated using the following expressions (Xi et al., 2016):

$$D_{ss,i} = 2d_{ss,i} = 2k_{ss} \times \sqrt{t_i} \tag{17}$$

$$R_{s,i}=\begin{cases} 100\%-\int_a^b \frac{\pi}{6} (D-D_{ss,i})^3/\int_a^b \frac{\pi}{6}D^3 \times 100\% & (a>D_s) \\ 100\%-\int_{D_0}^b \frac{\pi}{6}(D-D_{ss,i})^3/\int_a^b \frac{\pi}{6}D^3 \times 100\% & (a \leq D_{ss,i} \leq b) \\ 100\% & (b < D_s) \end{cases} \tag{18}$$

$$\Delta R_{s,i} = R_{s,i} - R_{s,i-1} \tag{19}$$

where $D_{ss,i}$ is the maximum diameter of SS that complete carbonation in the i-th year, $d_{ss,i}$ represents the depth of carbonation of SS in the i-th year, $k_{ss}$ is the rate of carbonation of SS, $t_i$ is the carbonation duration, D is the diameter of SS. $a$ and $b$ represent the corresponding minimum and maximum diameters of SS particles in a given size distribution. The annual carbonation of SS ($C_{ss,i}$) can then be calculated using the following expressions:

$$\varepsilon_{ss,i}=m_{s,i} \times r_{ss} \times f_{ss}^{CaO} \tag{20}$$

$$C_{ss,i} = \varepsilon_{ss,i} \times \Delta R_{s,i} \times f_{ss}^{use} \times \gamma_{ss} \times \frac{M_C}{M_{CaO}} \tag{21}$$

where $\varepsilon_{ss,i}$ is the mass of CaO in SS in the i-th year, $m_{s,i}$ represents the mass of crude steel that was produced in the i-th year, $r_{ss}$ is the output rate of SS, $f_{ss}^{CaO}$ is the concentration of CaO content in SS, $f_{ss}^{use}$ is the ratio of SS utilized as stacking and roadbed material and $\gamma_{ss}$ is the rate of conversion of CaO to $CaCO_3$ in SS.

## 2.4 Calculation of annual and cumulative uptakes

Even though the uptake of carbon can be estimated using alkaline materials in different stages of the lime cycle, the global and regional $CO_2$ absorption values were obtained via the aggregation of all alkaline materials. In global and regional carbon sink accounting, parameters such as the production of lime, proportion of lime utilized in different sectors, diffusion or carbonation coefficient, output rate, concentration of CaO, conversion ratio of CaO to $CaCO_3$, and particle size distribution and height of lime or red mud pile among others, were utilized as inputs for the model (see the SI-3 Data1). Basically, for the uptake of $CO_2$ in year $t_i$, the cumulative uptake of $CO_2$ in year $t_i$ minus that for year $t_{i-1}$ can be obtained from the following expression:

$$\Delta C_{l,total}^{t_i} = \sum C_{l,total}^{t_i} - \sum C_{l,total}^{t_{i-1}}$$

This allows contribution of the annual uptake of carbon to the total carbonation to be calculated.

## 2.5 Uncertainty analysis

We identified 16 groups of impact factors associated with the estimation of lime process carbon emission and carbon sequestration, which included 115 input-specific parameters, each with a specific statistical distribution (see the SI-3 Data1). Due to the difficulty in obtaining the true values of the parameters, we employed the Monte Carlo approach recommended by the 2006 IPCC Guidelines for National Greenhouse Gas Inventories to access the uncertainties for the carbon emission and removal of lime materials. We fed the statistical characteristics of the 115 variables into our models, and the simulated carbon emission and removal results were obtained through 10,000 iteration Monte Carlo simulation. Subsequently, statistical analysis was then performed to derive the median and the corresponding lower and upper bounds of the 95% confidence intervals (CI) for the carbon uptake and emission of lime materials.

## 3 Results

### 3.1 Aggregated regional and global emissions from the production of lime

The lime yield of various countries is shown in Figure 2a. To compensate for the underestimation of carbon sink and carbon emissions caused by the lack of data as much as possible (Cao et al., 2019), different data interpolation methods and parameters (as mentioned in the Section 2.1) were adopted to fit the lime output for 1930–1948, 1949–1962, and 1963–1995. The different interpolation methods and parameters led to changes in the uncertainty range, as shown in Figure 2a, which was reflected in the sudden change of data in the node years of piecewise fitting (such as 1948, 1949, 1963).

Considering the shortcomings of the global lime output statistics, this paper has made corresponding corrections to the global

lime output data based on China's lime output data (Fig. 2b). From 1930 to 2001, the cumulative value of compensated global lime production in this study is 7023.30 Mt from the missing data of China. Since 2001, the lime production in this study is a

slightly lower than that of USGS, due to the different reference sources of Chinese data. In general, the global lime output fluctuated and increased over time, from 139.62 Mt in 1930 to 394.93 Mt in 2019. In the early 1930s, the lime output decreased slightly, which may be due to the impact of the 'The Great Depression' and the closure of many factories, resulting in a decrease in the global lime output. In 2020, affected by the COVID-19, the lime production dropped slightly to 391.64 Mt (USGS: 427 Mt).

Figure 2c shows the estimated $CO_2$ emissions from lime production processes in China, the U.S., ROW and at a global scale from 1930 to 2020. According to our calculations, the global process $CO_2$ emissions increased from with 27.39 Mt C yr$^{-1}$ (95% Confident Interval, CI: 8.87–46.86 Mt C) in 1930 to 75.73 Mt C yr$^{-1}$ (95% CI: 69.18–82.33 Mt C) in 2020. In the early 1930s, carbon emissions slightly declined due to the impact of lime production and its uncertainty. The uncertainty of lime output can be transferred to the final simulation results of lime carbon emissions. The greater uncertainty of the parameters will lead to

greater uncertainty in the simulation results. The results of the 10,000 iteration Monte Carlo simulation show the change trend (Figure 2c). On a global scale, emissions doubled from 44.63 Mt C yr$^{-1}$ in 2002 to 75.73 Mt C yr$^{-1}$ in 2020. During this period (2002–2018), the average annual rate of increase was 2.98%, which was significantly higher than the rate for 1930–2002 (0.68%). The cumulative emissions of $CO_2$ from 1930 to 2020 were 3720.16 Mt C (95% CI: 3166.18–4287.43 Mt C). Emissions decreased in 2009, which was likely caused by the global financial crisis in 2008, during which downstream lime

industries experienced severe problems, such as excess produce, low production quantities, and stiff competition (Dong et al., 2010).

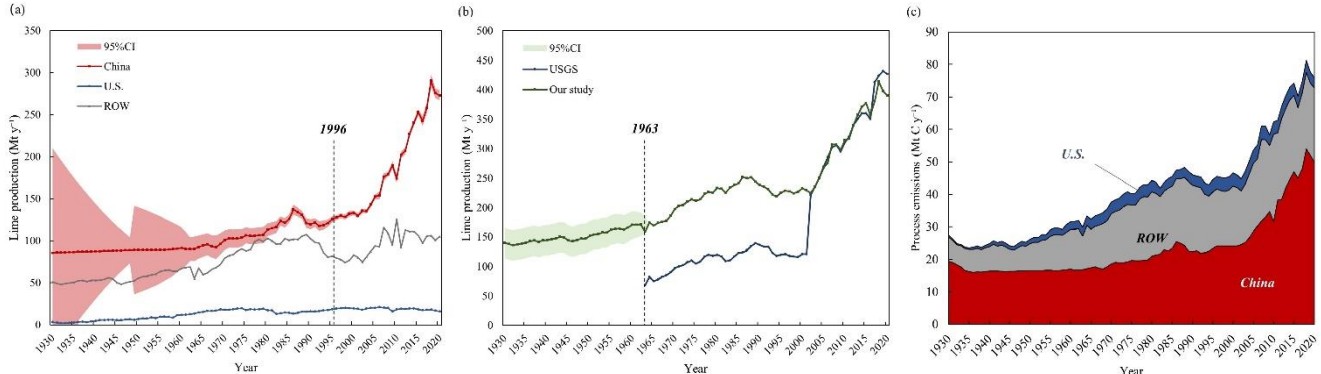

**Figure 2: (a) Lime production in different countries or regions from 1930 to 2020. Shadows represent uncertainty ranges. CI: Confidence interval. (b) Global lime production from 1930 to 2020. (c) Annual $CO_2$ emissions from industrial processes.**

$CO_2$ emissions from 1930 to 2020 in China account for approximately half of the global total. China was primarily responsible for the increase in the global emission from lime production processes during the studied period. In China, from 1930 to 2020, the average annual lime process $CO_2$ emission was 23.08 Mt C yr$^{-1}$, with 1.06% average annual growth rate. Notably, a rapid global increase in $CO_2$ emissions started in 2002. From 2002 to 2020, the average annual growth rate of carbon emissions from

lime was 4.03%, which was far higher than that of 1930 to 2001 (0.32%). This was mainly due to the steady growth of China's

macro economy after 2002. This finding was consistent with estimates from studies on the uptake of carbon by cement carbon based on similar approaches (Cui et al., 2019). These results are closely linked to the development of downstream sectors of the lime industry in China, such as the iron and steel, light and chemical, construction and materials industries (Shan et al., 2016b). In 2020, $CO_2$ emissions in China from lime production processes reached 49.93 Mt C $yr^{-1}$ (95% CI: 44.18–55.94 Mt C), and the cumulative emission was 2100.39 Mt C (95% CI: 1606.96–2620.93 Mt), accounting for 56.33% of the global total.

The current figure exceeds the 46.91 Mt C yr-1 forecasted for 2020 by Tong et al. (2019), which can be attributed primarily to the emission reduction scenarios they considered, assuming a technology penetration rate of 5% for CCU in China by 2020. However, it is important to note that as of 2020, CCU technology was seldom employed in China's lime industry. Therefore, the actual amount of carbon emissions produced by lime manufacturing is likely to be higher than in the scenario considered by Tong et al. Thus, our calculations are reasonable.

In the U.S., from 1930 to 2020, $CO_2$ emissions from lime production processes remained at around 2.72 Mt C $yr^{-1}$, and the cumulative emissions by 2020 were approximately 247.30 Mt C, which represents 6.63% of the global total. This relatively low value is because of a fairly stable production of lime in the U.S. and significant import of lime from Canada (USGS, 2022). Relatedly, for the ROW, the cumulative emission was 1380.77 Mt C, which represents 37.03% of the global total.

### 3.2 Lime uptake of carbon by regions

According to the lime carbon sequestration model, the global uptake of $CO_2$ by lime-containing materials increased from 9.16 Mt C (95% CI:1.84-18.76 Mt C) in 1930 to 34.84 Mt C (95% CI:23.50–49.81 Mt C) in 2020, representing an average annual growth rate of 1.50% (Fig 3a). Figure 3b shows the annual uptake of $CO_2$ in different regions, where the area represents the cumulative uptake in each region under natural conditions. In the early 1930s, the carbon sink of lime was affected by the uncertainty of lime production parameters, and the trend was slightly decreased, which was similar to the change of carbon

emissions in lime industrial process. Cumulatively, 1444.70 Mt C (95% CI:1016.24–1961.05 Mt C) were sequestered by lime-containing materials between 1930 and 2020. This means that 38.83% of $CO_2$ emissions from the production process of calcining limestone process were offset by lime carbon uptake at the same stage (1930-2020). The highest sequestration was in China (~63.95%, 918.41 Mt C) because of the associated high production of lime materials (China Statistical Yearbook, 2022), followed by the ROW (~34.35%, 474.35 Mt C) and US (~3.01%, 43.28 Mt C). China's lime carbon sink is greatly

affected by lime production, so its change is actually similar to that of lime production. The change of China's lime carbon sink was not obvious before the 20th century, fluctuating at 7.95 Mt C $yr^{-1}$. Until 2002, the total amount of carbon sink increased year by year with the increase of lime production. As seen in Fig. 3a, in China, lime carbon uptake increased from 10.52 Mt C in 2002 to 24.46 Mt C in 2020. Taking into account the data from 1930 to 2001 that we have fitted, we have compensated for the underestimation of China's lime carbon sink (cumulative 467.85 Mt C). Affected by the COVID-19, the amount of China's

lime carbon sink decreased in 2020 compared with that in 2019 (about 24.94 Mt C). For other regions, lime carbon sinks in

the United States (from 0.08 Mt C in 1930 to 0.66 Mt C in 2020) and the ROW (from 1.49 Mt C in 1930 to 9.24 Mt C in 2020) showed an overall trend of increasing over time.

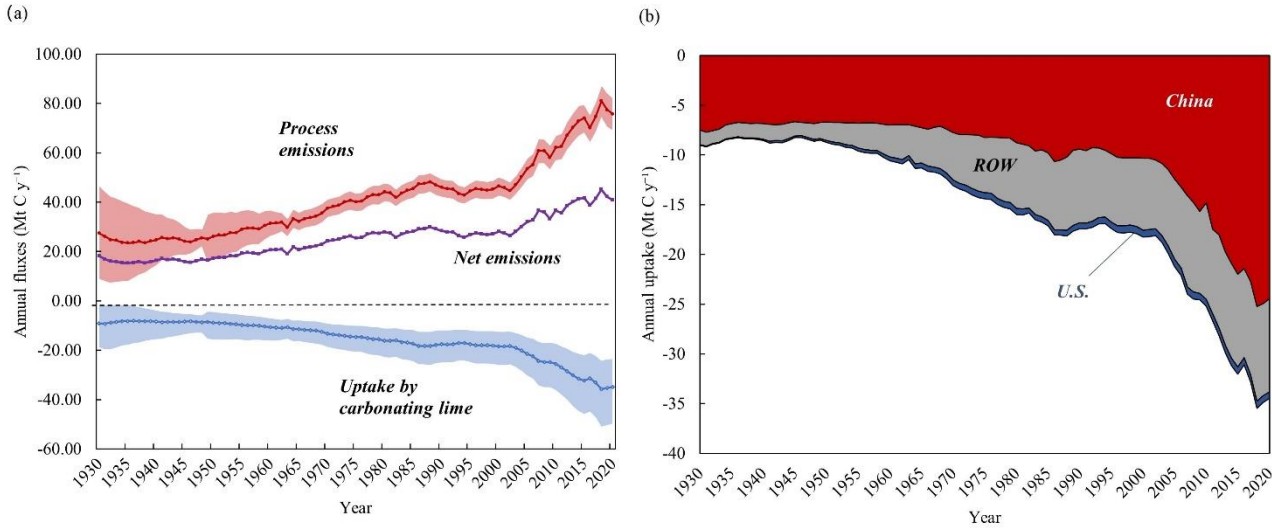

**Figure 3: (a) Net lime emissions from 1930 to 2020. Shadows represent uncertainty ranges. (b) Annual uptake of carbon dioxide by lime in different regions. ROW: Rest of World.**

The cumulative uptakes of $CO_2$ by lime materials in different regions are displayed in Fig. 4. Notably, the top three lime-containing materials (LSS, MOR and SS) accounted for 82.73% of the total global $CO_2$ uptake by lime. Regarding China, the cumulative uptake of $CO_2$ by all lime materials was 918.41 Mt C, and the amount of $CO_2$ that was removed by LSS (487.15 Mt C) exceeded the sum removed by all other materials. In the U.S., the uptake was dominated by carbonating SS, LSS, and SUG. The cumulative carbon sink of these three materials were 14.80, 7.26 and 6.69 Mt C, respectively. In the ROW, SS (175.72 Mt C), LSS (125.05 Mt C), and MOR (61.67 Mt C) were the top three materials.

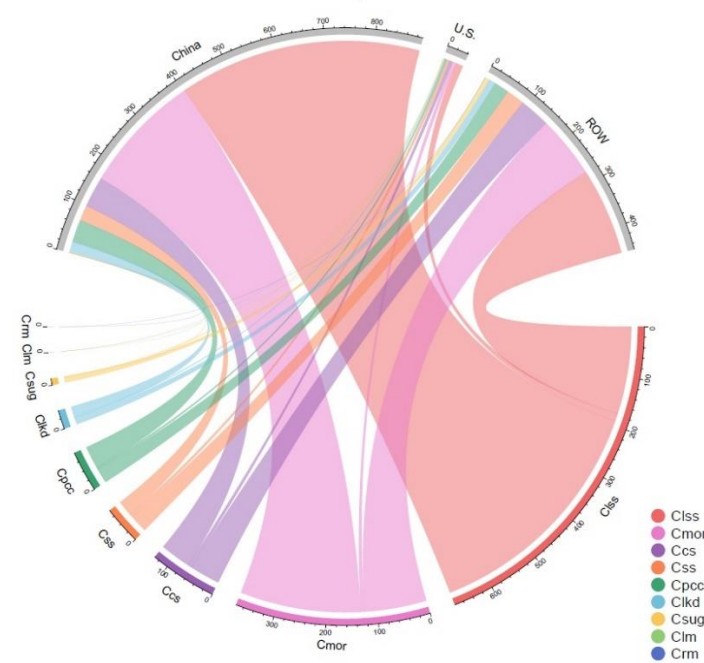

**Figure 4: Cumulative uptake of CO₂ uptake by lime-containing materials in different regions. ROW: Rest of World, Ccs: CO₂ uptake by carbide slag, Clkd: CO₂ uptake by lime kiln dust, Clss: CO₂ uptake by lime-stabilised soil, Cmor: CO₂ uptake by lime mortar, Cpcc: CO₂ uptake by Precipitated calcium carbonate, Crm: CO₂ uptake by red mud, Css: CO₂ uptake by steel slag, Csug: CO₂ uptake by carbonation sugar, Clm: CO₂ uptake by lime mud.**

### 3.3 Uptake of CO₂ in different stages of the lime cycle

Among the stages of the lime cycle, the service stage accounted for the highest uptake of $CO_2$ (1076.97 Mt C) from 1930 to 2020, representing 76.21% of the total. The uptake of $CO_2$ during the production and waste disposal stages were 36.95 and 299.19 Mt C, respectively (Fig. 5).

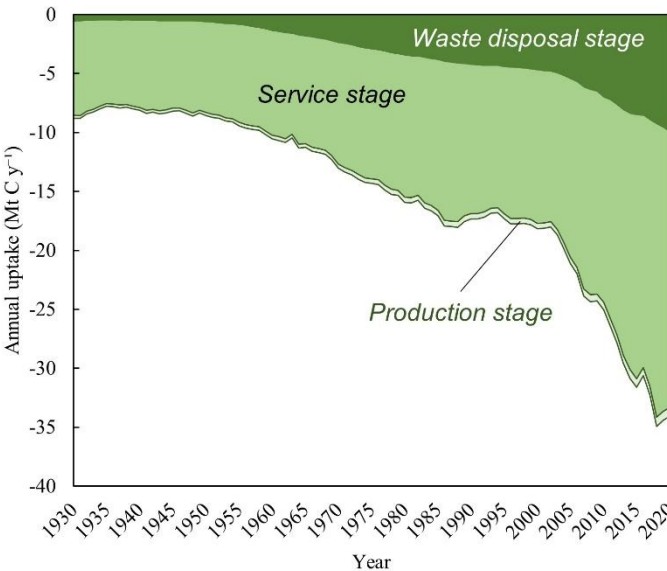

**Figure 5: Global annual uptake of carbon dioxide by lime in different stages of its cycle.**

Since 1930, the production stage is associated with a significant output of lime kiln dust (LKD), which is a by-product of the production of lime. The uptake of $CO_2$ by LKD in 2020 was 0.74 Mt C. This contribution is attributed to the development of

310 the lime industry and increase in the disposal of LKD in landfills (Latif et al., 2015). The concentration of CaO in the ash of lime kilns is approximately 54.88%, and thus, this continuously absorbs $CO_2$ in landfills (Bobicki et al., 2012).

The annual and cumulative uptake of carbon by lime materials during the service stage varied significantly, but these produced the following trend: LSS > MOR > PCC > SUG (Table 1). As commonly used building materials, LSS and MOR correspondingly removed 629.43 and 316.89 Mt C. Considering the consumption of lime in the construction sector over the

315 past five decades and its increasing utilization worldwide, especially in China and other developing countries, its uptake of $CO_2$ will certainly increase in the future (Renforth, 2019). The carbon fixation amounts of PCC and SUG of 84.98 and 45.68 Mt C, respectively, accounting for <10% of the total uptake during the utilization stage.

**Table 1. Summary of the global uptake of $CO_2$ by lime-containing materials in different stages of its cycle**

| Stage | Types of lime materials | $CO_2$ uptake in 2020 (Mt C) | Cumulative $CO_2$ uptake from 1930 to 2020 (Mt C) |
|---|---|---|---|
| Production | LKD | 0.76 | 36.95 |
| Service | LSS | 13.96 | 629.43 |
| | MOR | 6.88 | 316.89 |
| | PCC | 1.73 | 84.98 |
| | SUG | 0.74 | 45.68 |
| Waste disposal | RM | 0.002 | 0.05 |
| | SS | 8.31 | 225.67 |

| Stage | Types of lime materials | $CO_2$ uptake in 2020 (Mt C) | Cumulative $CO_2$ uptake from 1930 to 2020 (Mt C) |
|---|---|---|---|
| | CS | 1.67 | 73.39 |
| | LM | 0.003 | 0.09 |

**LKD: Lime Kiln Dust, LSS: Lime-Stabilized Soil, PCC: Precipitated Calcium Carbonate, SUG: Carbonation Sugar, RM: Red Mud, SS: Steel Slag, CS: Carbide Slag, LM: Lime Mud.**

Regarding the waste disposal stage, $CO_2$ absorption was mainly associated with carbonation of SS (Table 1). The cumulative uptake estimated in the present study was 225.67 Mt C. The iron and steel industry, which is a basic industry in industrialised countries, produces approximately 180–270 Mt of SS annually (Iron and Steel Slag, 2022). However, the alkaline content of SS is due to the high amount of lime used in the iron and steel making process. Therefore, SS sequesters a high amount of $CO_2$ in stockpiles and as roadbed material (Bobicki et al., 2012). Owing to its elevated concentration of $Ca(OH)_2$, high specific surface area and efficient carbonation process, CS is linked to the sequestration of approximately 73.39 Mt C (Huang et al., 2004; Hao et al., 2013). The total uptake of RM and LM is approximately 0.14 Mt C (Table 1). This low uptake is assigned to the high content of water in these wastes, which hinders the diffusion of $CO_2$ into their particles under exposure.

## 4. Discussion

Although the national greenhouse gas inventories guideline involves methods for quantifying $CO_2$ emissions that are linked to lime production processes, carbon sequestration of lime was not considered in the IPCC (IPCC, 2006). According to the analysis conducted in the present study, the uptake by lime-containing materials rapidly increased from 1930 to 2020 in all stages of the lime cycle. In 2020, the global uptake of $CO_2$ by lime was equivalent to 1.02% of the global industrial process emissions of $CO_2$; therefore, neglecting this sink caused an overestimation of the global carbon emission from industrial processes. The carbon sink increases over time, but this increase is due to an increase in production. It seems that both the increase in the sink and the emissions are proportional to each other. Our research results on carbon emissions and carbon absorption are significantly impacted by lime production. However, due to the lack of available data on annual lime production in China and worldwide during the early years, we used fitting methods to fill the gap of lime production and estimate it up to 1930. The statistically inferred 95% confidence interval was then used as the uncertainty range for lime production. To incorporate this uncertainty range into the accounting model for carbon sequestration and carbon emissions, we used Monte Carlo simulations, and after 10,000 iterations, we obtained the final accounting results for carbon sequestration and carbon emissions. Therefore, from the interpolation of production data to the final accounting of carbon sinks and carbon emissions, all potential sources of uncertainty have been fully considered in the accounting process. Thus, this is a crucial way to obtain lime carbon sink and carbon emissions data from 1930 to 2020 under current data conditions. However, as our understanding of basic data and the mechanisms of lime production, carbon sequestration, and carbon emissions deepens, and as we improve our activity level data, such as lime-based material utilization, waste stacking, and recycling rates, and optimize carbonization

parameters under different exposure conditions, there is still considerable potential for improving the accuracy of long time series lime material carbon sequestration and carbon emission accounting.

Regarding the global carbon cycle, lime's annual carbon uptake is estimated to be approximately 1.09% of the average global

land carbon sink from 2010 to 2020, which was approximately 3.18 Gt C yr$^{-1}$ (Global Carbon Budget, 2022). This indicates that lime's contribution to the global carbon cycle is significant and should be taken into account when considering strategies to mitigate carbon emissions. Therefore, if the lime sink is incorporated, the global carbon budget, which already includes data for carbon sinks of the ocean, land, and cement can be improved.

**Table 2. Comparison of $CO_2$ uptake by different types of materials**

| Region | Carbon sink type | Annual $CO_2$ uptake (Mt C yr$^{-1}$) | Source |
|---|---|---|---|
| Global | Carbonate | 660–1120 | (Li et al., 2018) |
| Global | Silicate | 34.64 | (Zhang et al., 2021) |
| Global | Lime | 23.50-49.81 | this study |
| Global | Cement | 207.27–291.82 | (Guo et al., 2021) |
| China | Steel slag | 1.36 | (Liu et al., 2018a) |
| China | Alkaline solid wastes | 10.91–30 | (Ma et al., 2022) |


To further illustrate the function of lime as a carbon sink, the results obtained in the present study were compared with data for the uptake of $CO_2$ by materials containing different minerals (Table 2). Rocks containing silicate and carbonate minerals are abundant in nature and are continuously extracting $CO_2$ from the atmosphere. According to recent studies, the annual average amounts of carbon sequestered by natural carbonate and silicate minerals are 890 and 34.64 Mt C yr$^{-1}$ (Li et al., 2018;

Zhang et al., 2021). However, the weathering of these minerals resulting in sequestration of $CO_2$ from the atmosphere occurs over a timescale of at least $10^4$ years (Berner et al., 1983).

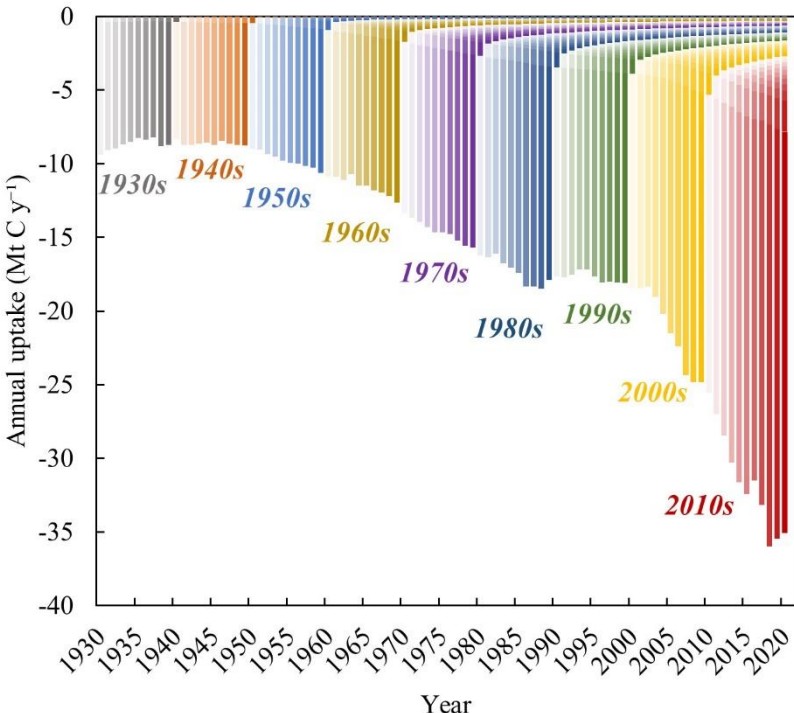

**Figure 6: Worldwide annual uptake of atmospheric CO$_2$ by lime, disaggregated by years of production**

Obviously, compared to natural carbonate and silicate minerals, the carbonation process involving alkaline materials produced by human activities, such as cement, SS and other solid wastes, is relatively faster under natural conditions (Berner et al., 1983). Lime materials, such as MOR and SS, similar to cement and natural materials, also remove of CO$_2$ from the atmosphere for several years or decades (Fig. 6). The uptake of CO$_2$ in each year includes lime materials that were generated or consumed in previous and current years: the former accounts for 15.59% of the total uptake, whereas the latter accounts for 84.41%. These results contrast with those obtained for the cement carbon sink, where most of the carbon absorption is linked to previous years. This difference is attributed to the higher calcium content, smaller particle size, and more active chemical properties of lime materials. These characteristics suggest that lime-containing materials, especially LKD and SS, are suitable for carbon capture and storage via mineralisation. Furthermore, conducting optimization studies on carbonization parameters under different exposure conditions and exploring the feasibility of employing CCUS (carbon capture, utilization, and storage) technology for lime-based materials could significantly advance research on lime carbon sequestration and mitigate impacts of CO$_2$ emissions (Pan et al., 2020).

## 5. Data availability

All the original datasets of CO$_2$ uptake by lime are available at https://doi.org/10.5281/zenodo.7759053 (Ma et al.,2023). This

dataset contains three data files, including lime material production and uses, lime carbon emission and uptake results, and the uncertainty of lime carbon emission and uptake.

SI-1 Lime carbon emission and uptake results, 1930-2020

Data 1. Annual carbon uptake by lime material and region

Data 2. Global carbon uptake by lime material and stage

Data 3. Global carbon uptake by region

Data 4. Annual global carbon uptake by lime material and relevant lag time, 1930 to 2020Data 5. Cumulative process

$CO_2$ emissions from lime production by region and category, 1930 to 2020

Data 6. Global process $CO_2$ emissions from lime production and carbon uptake by lime materials carbonation from 1930 to 2020

SI-2 Lime material production and uses, 1930-2020

Data 1. Lime production by region, 1930 to 2020

Data 2. Estimated production of lime in China, 1930 to 2020

Data 3. Estimated global lime production, 1930 to 2020

Data 4. Parameters of lime production fitting model

Data 5. Paper and paperboard production by region, 1930 to 2020

Data 6. Steel production by region, 1930 to 2020

Data 7. Alumina production by region, 1930 to 2020

Data 8. Output rate by material

Data 9. Estimates of lime used for different industries by region

SI-3 Uncertainty of lime carbon emission and uptake, 1930-2020

Data 1. Variables considered in the uptake uncertainty analysis using a Monte Carlo method

Data 2. The uncertainty of $CO_2$ emissions from lime production

Data 3. The uncertainty of lime carbon uptake

## 6. Conclusion

In the present study, a carbon sequestration model was utilized to quantify the global uptake of $CO_2$ by lime-containing materials from 1930 to 2020. The national greenhouse gas inventories guideline and global carbon budgets could be improved

by accounting for lime uptake, which can offset approximately 38% of emissions from industrial lime processes. The main findings of the present study are summarised below.

Global $CO_2$ uptake from lime production processes increased from 9.16 Mt C yr$^{-1}$ in 1930 to 35.27 Mt C yr$^{-1}$ in 2020. However, the cumulative uptake of $CO_2$ by lime-containing materials (1444.70 Mt C) offset approximately 38.83% of these emissions.

The uptake was highest in China (918.41 Mt C; 63.95% of global total) because of the associated elevated production and consumption of lime in recent decades. Uptake in the ROW and U.S. was 474.35 and 43.28 Mt C, respectively.

The uptake of $CO_2$ by lime-containing materials varied significantly at different stages of the lime cycle. In the utilisation stage, lime-containing materials, especially LSS and MOR, contributed the most to the total lime carbon sink (1076.97 Mt C). This was followed by sequestration in lime materials (mainly SS and CS) during the waste disposal stage (299.19 Mt C), whereas the production stage was associated with 36.95 Mt C. The sinks associated with the lime life cycle should not be neglected, and instead, they should be taken into account in future studies of the carbon cycle.

Historically, weathering of lime-containing materials was thought to occur over a large timescale. In the present study, it was revealed that approximately 15.59% of the annual uptake of $CO_2$ originated from lime that was produced in previous decades; therefore, this absorption potential cannot be ignored. In the future, carbon capture and storage can be improved via the use of lime-containing materials (e.g., SS and LKD).

**Author contributions.** LB and MM designed the study and prepared the manuscript with assistance from FX, JW, and LL. LL and MM performed the analyses, with the help of FX and LB on the analytical approaches. MM, LN, and FC performed the post-processing and analysis of the data as well as the review of the paper. LL and LB established the lime carbon sink accounting database, whereas LB and FX wrote the code and performed simulations of the datasets, with assistance from LL, MM, and LN. FX conceptualised and supervised the study.

**Competing interests.** The authors declare that they have no conflict of interest.

**Acknowledgements.** Longfei Bing, Mingjing Ma, and Fengming Xi acknowledge funding from the National Natural Science Foundation of China (No. 41977290), CAS President's International Fellowship Initiative (2017VCB0004), Youth Innovation Promotion Association, Chinese Academy of Sciences (grant nos. 2020201 and Y202050), Liaoning Xingliao Talents Project (No. XLYC1907148), and Natural Science Foundation of Liaoning Province (2021-MS-025).

**Financial support.** This research was funded by the National Natural Science Foundation of China (No. 41977290), CAS President's International Fellowship Initiative (2017VCB0004), Youth Innovation Promotion Association, Chinese Academy of Sciences (grant nos. 2020201 and Y202050), Liaoning Xingliao Talents Project (No. XLYC1907148), and Natural Science Foundation of Liaoning Province (2021-MS-025).

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
