# Peer review of "An investigation of the global uptake of CO2 by lime from 1930 to 2020"

_Earth System Science Data, 2022_

## Referee Comment (RC1)

Thanks to authors for intent to share data. This reader can easily download and open any of three products from Zenodo.

Not sure how to react to this product? Good methods but trivial outcome? Absence of necessary information and incompatibilities with other products cloud my overall judgement.

Serious questions first, followed by technical issues.

Time period. Title implies 1963 to 2020. Authors never explain why 1963 nor why 2020? Guo et al. (in ESSD 2021: https://doi.org/10.5194/essd-13-1791-2021) report - for cement - 1930 to 2019. Andrew, also considering cement but published a year earlier (https://doi.org/10.5194/essd-11-1675-2019) used 1928 to 2018. Global carbon budget (multiple versions published in ESSD but here I use GCB 2021, https://doi.org/10.5194/essd-14-1917-2022, to extract data for 2020 used below) covers (in detail) 1959 through current year. Although these authors expend considerable text (e.g. in lines 69 to 83) to justify closing gap in China data (evidently restricted to 2002-2016) via statistical extrapolation, readers never see explanation for start year of 1963. Comes from US, via USGS? Or something from RoW? Readers can guess why data extend to 2020, but 1963 never explained and doesn't match other products.

Problem with single years. Authors discuss 2020 lime emissions but with almost no discussion: a) they had to extrapolate - at least for China data - to get to that number; b) we know that any single year carries very large (larger than 95% CI) uncertainty; and c) we know that 2020 particularly, occurring as it did as countries emerged (or, not) from economic and social impacts of pandemic, itself represented an unusual year.

Impact estimate. Weaknesses in using specific single years notwithstanding, this reader evaluated 2021 GCB (https://doi.org/10.5194/essd-14-1917-2022). I find total 2020 emissions as 9.3 $\pm$ 0.5 GtC/yr (using customary well-justified GCB $\pm$ 5% uncertainty) when they include cement carbonation correction. Without carbonation correction they report 9.5 GtC/yr, so total cement carbonation term = 0.2 GtC/yr in 2020. Authors of this manuscript give 134.33 Mt CO2, converts to approximately 36 MtC per 2020. Given overall uncertainties of 500 MtC per year and total cement carbonation of 200 MtC per year in 2020, numbers presented here (36 / 200) represent at most 20% correction on carbonation (numbers provided by these authors in their Table 2 tend to support my estimates) and only 0.4% correction on overall emissions, well within uncertainty limits. Not clear to this reader why authors consider this work worthy of ESSD publication? Although they proclaim (lines 22-23) importance of lime processes in carbon cycle, actual numbers (which they never quote but should) prove otherwise. On the other hand, we need to quantify all these corrections (however minor) so authors need to demonstrate special efforts and skills to certify extremely small numbers. Not clear to this reader how authors came up with 38% number (line 19) nor what "associated processes" (line 20) they refer to. Later they quote another author (line 42) that total lime-related carbonation occurs within only 34% of lime produced. Authors' obvious enthusiasm for their topic obscures practical impact; this reader had to resort to own calculation (above). Most readers will need a statement about minimal current impact, coupled with justification to quantify as precise albeit small correction and as possible future mitigation option.

Please can I also suggest that: a) these authors need to reconsider how they can justify reporting unrealistic uncertainties at tenths or hundredth of Mt CO2); and b) work could have stronger impact if they adopted units (C rather than CO2) and uncertainty conventions of GCB.

ESSD references. It seems poor form to miss so much highly-relevant prior work, much of it cited above, published in the journal which these authors choose to use. Perhaps with focus on lime processes and lime engineering they have missed relevant climate impact literature?

Fair enough if they want engineers as their audience but they clearly reach for climate relevance. If so, they need to cite relevant literature.

Posting uncertainties in separate file to actual data seems unusual at best? Derives from authors' mechanisms (after-the-fact Monte Carlo simulations) to estimate uncertainties? Validations very challenging for one-off products but I have given some budget-related hints above. Authors might know other validation options? Unfortunately, these authors never address validation. Perhaps they should review ESSD guidelines at https://www.earth-syst-sci-data.net/10/2275/2018/. I decided not to list further line-by-line technical changes. I vacillate between recommending rejection or recommending major revisions; I leave that question to handling editor. In reading this manuscript I wanted but could not find key justification and relevance nor acceptable uncertainties nor any validation attempt.

---

## Referee Comment (RC2)

**Review of Bing et al., 2022**

This study analyzes the global and regional uptake of $CO_2$ by lime through the different process of lime production from 1963 to 2020. This study is of interest for the global carbon community, as it is important to more accurately account for sources and sinks of $CO_2$ by lime-containing materials for better estimation of its impact on the carbon cycle. However, the manuscript is not clear for certain aspects of the study, a lot of information is missing, and more analyses should be done. I could not find the supplementary information, and the study mentions a model to conduct this work but no information on this model is provided in the manuscript. Please, find my comments below.

**General comments:**

Ln. 22. For the period 1963 to 2020, your results show that 38.79% of $CO_2$ emissions were sequestered by lime production from the total global $CO_2$ lime emissions. So, your results and this sentence suggest that global $CO_2$ sink from lime corresponds to ~40% with ~60% of global $CO_2$ source from lime. How, Ln 22-23, can you justify that lime materials is a global carbon sink that can reduce the carbon footprint of lime production? Need more clarification in the abstract.
Additionally, in the abstract you are only giving numbers of the $CO_2$ lime uptake at a global scale and for China. What about the $CO_2$ lime emissions for the 1963-2020 period for China and at global scale?

Ln 65. You mentioned "this study significantly improves the global carbon uptake model" but what was improved and how? No information in the whole manuscript could be found on this improvement. More information should be provided on this improvement.

Ln.80. You assumed for the PCA calculation that you have a linear relationship between variables, but how would you justify that there is a linear relationship for the data over the period 1963-2000? Additionally, is the lime production linear over the period 1963-2020 for all three regions?

Ln. 96. "Considering the availability of lime production data", what do you specifically mean by availability of the data? Additionally, what are the uncertainties associated with the dataset used in this study? Any uncertainty in the lime production data of China and US?

Ln 96 and Equation 1. Why are you using method 1 and not method 2 (which use a correction factor) or 3 here? Should be mentioned in the text. The indexes "l" and "i" in your equation are not defined.

Ln 101. Is this method for estimating emissions factors based on tier1 method from IPCC 2006? If yes, which tier method for choosing the emissions factor are you using exactly?
Emission factor for combustion have values depending on the category and on the target year. Are the values of emission factor used for lime production and in this line for specific year and specific process type?
Based on the table available from IPCC 2006 (https://www.ipcc-nggip.iges.or.jp/EFDB/find_ef.php?ipcc_code=2.A.2&ipcc_level=2), developing countries emission factor for dolomitic lime production is equal to 0.77 t $CO_2$/tlime while developed countries value is of 0.86. The emission factors mentioned in your paper do not correspond to those from IPCC database, can you develop why?

What are the uncertainties associated with the emission factor and how did you count of these uncertainties in your study?

Equation 3. According to IPCC 2006, the amount of dust depends on the type of kiln used in lime production. Have you considered the type of kiln in the parameter $r_{lkd}$ of your equation?

Section 2.3. In this section, there is no information on where all parameters used for the equation come from. For instance, there is no information on the rate of Cao to $CaCO_3$ in dust. More information on these parameters should be inserted.

Ln. 193. "[…] were utilized as inputs for the model (see the Supplementary Information)." First, I could not find the Supplementary Information. Second, there was no information of the model used for this study so far. Model name, model goal, model algorithm, … were not introduced so far. There should be a section to describe the model used for this study.

Ln. 195. This expression is not clear. Why use the year ti-1 to calculate the uptake of $CO_2$ in year ti? It is not clear also how the contribution of annual uptake of carbon to the total carbonation can be calculated by using this expression. Please, clarify.

Section 2.5. The uncertainty analysis is not enough detailed. The carbon absorption factors, and activity level data are only introduced here. More information on these parameters are needed. More information on the model uncertainty and statistics should be found here and not in supplement information (which cannot be found).
As mentioned in IPCC 2006, complete activity data are needed. Omission of use or lime production as a non-marketed intermediate, not well accounted for in inventories, may lead to an underestimation of lime production by country by a factor of 2 or more. Uncertainties associated with LKF are also non negligeable. Have you accounted for these uncertainties by country?

A section should give information on the carbon sequestration analytical model used for this work.

Ln.207-211. You mentioned a decrease in emissions in 2009 due to the financial crisis of 2008. But we can also observe a decrease in 2017, 2018, and some other years before. What are the reasons for all these decreased emissions?
Additionally, what is the reason for the sudden increase in emissions starting 2002? Ln. 213 seems to answer this question, but it is not clear if the increase mentioned Ln. 213 refers to the large increase starting in 2002.
You should give a description of the net emissions here. As we can see, net emissions (process + uptake) show net source of $CO_2$ emission at global scale.
It is not clear also if these results are based on the inventories or your model results. Please, clarify.
"Subtraction of the amount of $CO_2$ absorbed from $CO_2$ emissions", it is not clear which $CO_2$ absorbed from $CO_2$ emissions are you talking about. Please, clarify.

Ln 214. There should be information as well about the period 2002-2020 for comparison with the global annual average.

You considered ROW as regions of developing countries Ln. 101. However, countries considered as developed countries might be included in it, such as Canada. Ln. 225, you mentioned there is "significant

import of lime from Canada" to the US. Is Canada part of ROW region? If yes, can you really consider ROW as developing countries?
If ROW region was grouped into two categories (developed and developing countries), what would be the $CO_2$ emissions for both categories?

Ln. 231. What is the percentage of uptake of $CO_2$ compared to the emissions of $CO_2$ at global and regional scales?

Figure 3.b. Results from this figure have not been used. What is this figure used for and telling us?
Ln. 247. Should be Fig.3b

Ln. 248. This sentence is not clear. Do you mean, the uptake of $CO_2$ by dust decreases or increases since 1963? Uptake of $CO_2$ by dust in 2020 is available but value for 1963 should be mentioned here as well.

Ln. 270. According to your results, "the uptake of lime-containing materials rapidly increased from 1963 to 2020", but you mentioned earlier as well that this increase is proportional to the lime-containing materials uses and production. Does the uptake increase proportional to the emission during the period, or is this uptake increasing at a certain point? Maybe your results should be displayed by removing the trends, or additional detrending results should show the uptake and emissions of $CO_2$ during all stages of the lime cycle, by region and a global scale. With your current results, it is difficult to say if there is or is not an increase in $CO_2$ uptake by lime-containing materials.

Ln. 275. You mentioned that if the lime sink would be incorporated in the global carbon budget (GCB), the carbon sinks could be improved, however, let's not forget that the emissions of $CO_2$ associated with the calcination of lime and limestone outside of cement production is not considered in the GCB. Knowing that the net emissions from lime production are a source of $CO_2$, if the lime source and sink were included in the GCB, how much do you think the net emission from the GCB would change? Discussion on this aspect should be included here.

Ln. 285 and Figure 5. Additional results should show the cumulative sources of $CO_2$ as well as the cumulative net $CO_2$ emissions. And further discussion should be added about the net emissions.

Ln. 300. How can we consider lime as a carbon sink when the net emissions show a source of $CO_2$?

**Specific Comments:**

Ln. 14. Which existing data are you referring to? You are analyzing these data (satellite, in situ?) in your study, so you should mention here the data used (names of data?).
This sentence contains twice materials ("materials associated with the production [...] of lime-containing materials"), could you rephrase?

Ln. 15. The model used should be mentioned here.

Ln 20. Which associated process are you mentioning? Is this process the production of lime materials?

Ln. "Total uptake", do you mean total global uptake?

Ln 62. The three stages should be mentioned here for clarification:

Limestone calcination (lime production)
Hydration reaction (lime decomposition)
Lime carbonation

Ln. 72. "[…] data on lime production from 1963 to 2000 in China were not available in the existing databases." What do you mean? Regarding the following sentence, you should mention that there was no data during this period in the China Statistical Yearbook.

Ln. 79. What are xxx and xx? Need more precision. "A linear regression was then built with xx", what does xx correspond to?

Ln. 94-96. The sentence needs to be rephrased.

Ln.98. "in the present", what do you mean by it? Today?

Ln. 105. A reference is missing here.

Equation 2. what is "l' referring to?

Ln. 117. Units are missing

Ln. 155. How many years are sufficient consequently?

Equation 11. Indexes "m" and "p" are not defined.

Ln. 182. Only "a" and "b" are in equation 18. Where are "$a$" and "$b$"?

Ln. 193. "were utilised" should be "were utilized"

Figure 1 was not used in the manuscript.

Ln. 204. "and at a global scale" should also be mentioned.

Ln. 445. (a) should mention "at a global scale"

Figure 2. Are the global annual $CO_2$ emissions from IPCC 2006? The dataset used for US, China and ROW emissions were defined in the methodology but not the global annual emissions.

Figure 2.a. What are the shadows representing? Information on this shadow should be included in the caption of the figure.

Ln. 232. "whereas the area represents the cumulative uptake in each region under natural conditions." it is not clear which area you are talking about. Which natural conditions are you mentioning? For which process?

Ln. 205. The meaning of CI (Confidence Interval) should be mentioned in the text.

Ln. 233. Is this value for global scale or for a specific region?

Ln. 273-275. Specific values in Gt/yr should be mentioned here.

Ln. 288. "inconsistent", what do you mean by these results are inconsistent with those from cement carbon sink? Maybe inconsistent should be changed with another word: "these results contrast with those obtained for the cement carbon sink […]"?

Ln. 294. CO2 -> $CO_2$

Ln. 301-302. Global $CO_2$ emissions reported here in $MtCO_2.yr^{-1}$ are smaller than reported for China in section 3.1. Please, revise.

---

## Author Comment (AC3)

**In response to 'SUMMARY and MAIN IMPRESSIONS':** Thank you for recognizing the importance of this piece of research. Regarding the two issues you mentioned, on the former, we have presented the detailed information, including activity level data and carbon absorption factor data, as well as uncertainty analysis results by the Monte Carlo simulations, in the dataset at https://doi.org/10.5281/zenodo.7112485. And we will further improve the data information and description in the next revision. On the latter, according to the diffusion characteristics of $CO_2$ in different types of lime materials, we used the method of greenhouse gas inventory analysis, combined with Fick's second law, to build the mathematical model of lime carbon sink. See formula 2-21 for details. Furthermore, we will address your concerns in the following texts.

**In response to 'GNERAL COMMENTS':**

Abstract: Thank you for your suggestion. Lime is obtained by calcining limestone at high temperature, which will also produce by-product lime kiln ash. Globally, lime is mainly used in chemical industry, environmental protection, metallurgy, construction and other industries, and it can also be used to produce refractory materials. During use and waste disposal, lime will react with water and carbon dioxide in the air to produce calcium carbonate. According to the definition of "carbon sink" by IPCC, the carbon absorption of lime during use and waste disposal belongs to carbon sink, but has been ignored. We will add the clarification of lime carbon sink mechanism mentioned above in the revised abstract. As for another comment,

the focus of this study is on the carbon sequestration accounting of lime. Therefore, more emphasis is placed on the carbon absorption of lime in this period. For clarification, we simply emphasized the proportion of lime carbon sink to lime carbon emission.

Ln 65. We are grateful for the suggestion. To be more clearly and in accordance with the reviewer concerns, we have added a more detailed interpretation as following: Due to the lack of accounting methods and the difficulty in obtaining data, the lime carbon sink has not been revealed. According to our calculation, the annual carbon sequestration of lime accounts for about 1.03% of the global terrestrial carbon sequestration. In addition to cement and alkaline solid waste, lime carbon absorption is also a carbon sink generated by human activities. The research results can explain the whereabouts of some missing carbon sinks in the global carbon sink, and then make up for the global carbon sink model.

Ln 79. We have re-fitted China's lime production. And relevant expressions have also been readjusted. The USGS records China's lime production from 1990 to 2020, but they are all estimates. In China, the only record of China's lime production is the "China Building Materials Industry Yearbook", whose values are provided by the China Lime Association. Its dataset is shown in Figure 1 for actual production, covering the years 1996-2014 and 2019-2020. Considering that the downstream sectors of lime in China mainly include construction, steel, calcium carbide, and alumina, we collected China's cement production (1930-2020), total housing area completed (1963-2020), steel production (1949-2020), calcium carbide production (1949-2020), and alumina production (1954-2020) as variables to fit the lime data. We have adopted the multivariate linear regression method you suggested. In order to avoid multicollinearity, a stepwise method is used to establish a model. The dependent variable in the model is lime yield, and the independent variable is calcium carbide yield, the completed area of houses in the whole society, cement yield and alumina yield. The determination coefficient of the model is 0.9954, and the adjusted determination coefficient is 0.9942. Given that data on

the completed area of housing in the whole society is limited to after 1963, the model only predicts lime production data from 1963-1995 and 2015-2018 (that is, the red line in Figure 1, the shaded part is the 95% confidence interval of the estimation result).

Ln 80. Thanks very much for reviewer's questions.

Data of lime production in 2001-2016 is compiled by the Chinese Lime Association, the only institution that records the production, and their figures are based on the demand of downstream consumer sectors, including crude steel, construction and chemical industry. Among them, the amount of lime used in crude steel, calcium carbide, construction and alumina accounts for more than 90% of the total lime production. Therefore, we select crude steel production, calcium carbide production, housing completion area and alumina production as the four variables for predicting lime production. Although there are no specific annual data records from 1963 to 2000, the use of lime in China has a long history. From the 7th century BC, lime became an important building material in China and began to be used, and according to the Chinese Statistical Yearbook, from 1963 to 2000, China was produced with calcium carbide, crude steel, alumina and other industrial products. Therefore, we assume that the data of China's lime production from 1963 to 2000 are still related to these four major industries. Your question is very reasonable. We also used autoregressive models and other methods to predict the data from 1963 to 2000, but the method mentioned in this article obtained the largest coefficient of determination.

For the other three regions, we have carried out the same analysis for the United States, the world and other countries. According to USGS records, the downstream use sectors of lime in the United States mainly include construction, chemical industry, metallurgy, environmental protection and refractory materials. We collected the crude steel production, alumina production and cement production in the United States from 1963 to 2020. The correlation between each variable is shown in the table below. It can be seen that there is a strong correlation between the lime production in the United States and each variable. In the construction of a general linear model, the determining coefficient of the model is 0.636.

**Table 1** Correlation of various variables in the United States from 1963 to 2020

| | crude steel production | alumina production | cement production | lime production |
|---|---|---|---|---|
| crude steel production | 1 | | | |
| alumina production | 0.827** | 1 | | |
| cement production | -0.017 | -0.036 | 1 | |
| lime production | 0.335* | 0.289* | 0.716** | 1 |

\*\*. Correlation is significant at the 0.01 level (2-tailed).

\*. Correlation is significant at the 0.05 level (2-tailed).

Likewise, for other countries, there is a strong positive correlation between lime yield and other variables. A general linear model is constructed with lime yield as the dependent variable and

other variables as the independent variable, and its determination coefficient is 0.749.

**Table 2** Correlation of variables in ROW

|  | crude steel production | alumina production | cement production | lime production |
|---|---|---|---|---|
| crude steel production | 1 |  |  |  |
| alumina production | 0.942** | 1 |  |  |
| cement production | 0.903** | 0.951** | 1 |  |
| lime production | 0.782** | 0.613** | 0.602** | 1 |

** Correlation is significant at the 0.01 level (2-tailed).

From 1963 to 2020, the global lime production has a strong positive correlation with the global crude steel, alumina, and cement production. Taking the global lime production as the dependent variable and other variables as independent variables, the obtained model has strong collinearity. Therefore, the principal component analysis method is selected to obtain a principal component, and the coefficient of determination obtained by the model is 0.946.

**Table 3** Correlation matrix between global lime production and various variables

|  | crude steel production | alumina production | cement production | lime production |
|---|---|---|---|---|
| crude steel production | 1 |  |  |  |
| alumina production | 0.984** | 1 |  |  |
| cement production | 0.985** | 0.986** | 1 |  |
| lime production | 0.980** | 0.964** | 0.959** | 1 |

** Correlation is significant at the 0.01 level (2-tailed).

Therefore, this linear relationship is also very obvious on a global scale, but this linear relationship is relatively weak in the United States and other countries.

Ln 96. According to the USGS, the data of global lime production began in 1963. And the " availability of lime production data " mentioned in this paper refers to the data currently available to statistical institutions or literature journals. There are many uncertain detailed data, which are included in the supplementary materials.

Ln 96. and Equation 1. We are very grateful to the reviewer for this comment. As we mentioned above, Tier 1 is based on the default emission factor of the national lime production process provided by the IPCC, and the proportion of usage of various lime uses; the emission factor in tier 2 is collected by national or plant level data; for Tier 3, the emission factor should be specific to the carbonate consumption process in each factory. The lime production data in this study belong to the national level, and the emission factor uses the default emission factor provided by the IPCC, which is not detailed to the factory level of each country, so Tier 1 method is selected; Thanks very much for reviewer's opinion. In Equation 1, l refers to different

types of lime use, including PCC, sugar making, lime-stabilized soil and lime mortar, and i refers to different years. We will add it in the revision.

Ln 101. I'm very sorry for the careless typo here. The emission factors used in our actual calculation are 0.683 for China, 0.77 for the United States, and 0.75 for other countries. Among them, we use IPCC2006 tier1 for the emission factors of the United States and other countries, and China's emission factor is based on China's provincial greenhouse gas emission guidelines, which is consistent with the research of Shan and is more in line with China's reality.

In the IPCC guidelines, Equation 2.8 illustrates how to calculate the Tier 1 emission factor for lime production.

*EQUATION*2.8

$$EF_{lime} = 0.85 \times EF_{high\ calcium\ lime} + 0.15 \times EF_{dolomittic\ lime}$$

the EF $_{high\ calcium\ lime}$ is 0.75, and the default EF for dolomitic may be 0.86 or 0.77 for developed countries and developing ones. Substituting it into the formula, the emission factors of developed and developing countries are obtained as 0.77 and 0.75, respectively.

When setting the uncertainty of emission factors for different countries, we considered the emission factors for all lime types mentioned in the IPCC guidelines. The minimum value of the emission factor is 0.59, that is, the hydraulic lime type, and the maximum emission factor is 0.86 for developed countries. For developing countries, it is 0.77. Therefore, in our model, the maximum and minimum emission factors for China and ROW are 0.77 and 0.59, respectively, and 0.86 and 0.59 for the United States.

Equation 3. The ash yield of different lime kilns is mainly obtained through literature collection (the ash yield of lime kilns is between 0.09 and 0.1). As an important calculation parameter of lime kiln ash, it participates in the simulation of 100000 times by Monte Carlo method.

Section 2.3. Thank you for your suggestion. Because there are many parameters involved in this step, we put the relevant reference sources in the SI file, and the link is https://doi.org/10.5281/zenodo.7112485. We are happy to add this part in the next revision

Ln193. We can agree on this. The lime carbon sink model here is constructed according to the diffusion law of $CO_2$ in different lime materials. For the carbon sink model of each material, see the public trial 2-21. The Ln 193 part represents the accounting method of the annual lime carbon sink. The description of the carbon sink will also be added in the next revision.

Ln195. Because different lime materials have different carbon absorption rates, some alkaline solid wastes with large particle size cannot be completely carbonized within one year. Therefore, to calculate the carbon sink of lime materials in the current year, the accumulated carbon sink of lime in $t_i$ year should be subtracted from the carbon sink of lime in $t_{i-1}$ year. In the next version, we will also add a description of this part.

Section 2.5. The input variables and parameter distribution of uncertainty are shown in the annex. For your suggestions, we will also add relevant descriptions of uncertainty in the next version

Section 2.5. Uncertainty analysis description and carbon sequestration analytical model will be further improved in the next manuscript. In response to the question about

Ln.207-211. (1)Reasons for the decline in other years: In addition to 2008, other significant carbon emissions decline years include 1982, 1990, 1993, 2002 and 2016.

(2)China's lime carbon emissions have risen sharply since 2002, with an average annual growth rate of 4.86% from 2002 to 2020, compared with an average annual growth rate of 0.65% from 1963 to 2002. This is mainly due to the steady growth of China's macro economy after 2002.

(3)The results in Ln 204-211 of the article are based on our model's calculations, which we will describe in a revised draft thanks to the expert's reminder.

(4)Thank the experts for their suggestions on net emissions, and we will add this part to the revised version.

Ln.214. Thanks to the advice of reviewer, the average annual growth rate of China's lime carbon emissions from 2002 to 2020 was 4.86%, which was higher than the global average annual growth rate of 3.69% during the same period.

Ln 231. According to bp 'bp Statistical Review of World Energy', The total global carbon dioxide emissions in 2020 is 32.284 billion tons. In 2020, the total global carbon dioxide emissions will be 32.284 billion tons. In the same year, the carbon emission of lime industry process was about 293.98 million tons, accounting for 0.91% of the global total carbon emissions. For different regions, China, the United States and other countries account for 2.13%, 20% and 30% of the total carbon emissions of the lime industry respectively

Figure 3.b. mainly shows the carbon sequestration of lime in different life cycle stages. We have revised 'Figure 2.b' to 'Figure 3.b' in Ln 247.

Ln. 248. Well, the carbon absorption of dust is increasing. We will introduce the carbon absorption in 1963 to further improve the description.

Ln. 270. Lime carbon sink and industrial process emissions show an upward trend from the trend. We show the change trend of emissions and absorption in different regions in Figure 2b and Figure 3a. The carbon emission of lime refers to the emission of limestone calcination production, which is the carbon emission of industrial process. In Figure 3b, we show the carbon absorption in the global life cycle. The carbon sink results calculated by us increase with time, which can explain this situation.

Ln. 285 and Figure 5. the cumulative sources of $CO_2$ as well as the cumulative net $CO_2$ emissions will be showed in the next revised draft. And we will add further discussion about the net emissions.

Ln. 300. In fact, our logic is: In our study lime carbon sink has been revealed, which is one of the important destinations of carbon emissions in the lime industry process, in other words, some industrial process carbon emissions are absorbed back by lime. If carbon sink is considered in the carbon emission accounting of industrial process, we can determine the net

carbon emission, then we have solved the problem that the lime industrial process is overestimated.

**In response to 'SPECIFIC COMMENTS':**

Ln. 14. The existing data refer to the data available in the national statistical yearbook, literature journals, international databases, etc. This part of data is included in the annex, and we will display it in the text according to your suggestions. We rephrased Ln14-16: Here, existing data on materials associated with the production, utilisation, and disposal stages of lime-containing materials were analysed using lime carbon sink accounting model to obtain regional and global estimates for the sequestration of carbon from 1963 to 2020.

Ln 15. The model name is 'lime carbon sink accounting model'.

Ln 20. Yes. This process refers to the industrial process of lime carbon emission. We will improve the description of this part.

Ln. Yes. Total global uptake.

Ln 62. Agree with your opinion. These three parts will be supplemented.

Ln. 72. Yes, your statement is more precise. 'There was no data during this period in the China Statistical Yearbook.'

Equation 2. 'l' in Equation 2. refers to lime

Ln 79. We have revised the estimation of lime production simulation. We have adopted the multivariate linear regression method. In order to avoid multicollinearity, a stepwise method is used to establish a model.

Ln 94-96. Rewritten sentence: Lime comes from the decomposition of limestone in shaft kiln or rotary kiln, and the carbon emission of this industrial process is estimated from the IPCC method (IPCC, 2006).

Ln 98. It refers to the current stage, or today.

Ln 105. Thanks to the advice of reviewer, The references here is "PR China National Development and Reform Commision. Guildelines for provincial greenhouse gas inventories; 2011. [Chinese Document]." We will add it to the revised draft.

Ln. 117. The problem has been corrected. Unit: $g/mol$

Ln. 155. It usually takes several years or even decades.

Ln 204 Thanks very much for reviewer's reminder, we will add explanations on a global scale.

Ln. 445. This part is changed to: Annual CO2 emissions from industrial processes and the associated uptake by lime from 1963 445 to 2020 at a global scale.

Equation 11. Indexes 'm' means red mud or lime mud; 'p' means production.

Ln. 182. In Equation 18.

Ln. 193. Change 'utilised' to 'utilized'

Figure 1. mainly shows the situation of different life cycle stages of lime

Ln. 445. OK.

Figure 2. In fact, the global annual carbon emissions of lime come from our calculations. And we will make this part easier to understand.

Figure 2.a. Shadows represent uncertainty ranges.

Ln. 232 The areas with different colors represent the accumulated value of lime carbon sink in different areas.

Ln. 205. Thanks very much for reviewer's reminder, we will add the meaning of CI (Confidence

Interval).

Ln. 233. 4053.61 Mt of $CO_2$ is for global scale.

Ln. 273-275. Specific values in Gt/yr will be mentioned in the next revision.

Ln. 288. Thank you for your modification. Your statement is more scientific and rigorous

Ln. 294. Chang 'CO2' to '$CO_2$'.

Ln. 301-302. Change 'emission' to 'absorption' in line 301.

---

## Author Response (AR1)

**RC1**

Thank you for your precious comments and suggestions. Those comments are all valuable and very helpful for revising and improving our paper, as well as the important guiding significance to our researches. The responds to the reviewer's comments are as following:

**1.Comments**: Time period. Title implies 1963 to 2020. Authors never explain why 1963 nor why 2020? Guo et al. (in ESSD 2021: https://doi.org/10.5194/essd-13-1791-2021) report - for cement - 1930 to 2019. Andrew, also considering cement but published a year earlier (https://doi.org/10.5194/ essd-11-1675-2019) used 1928 to 2018. Global carbon budget (multiple versions published in ESSD but here I use GCB 2021, https://doi.org/10.5194/essd-14-1917-2022, to extract data for 2020 used below) covers (in detail) 1959 through current year. Although these authors expend considerable text (e.g. in lines 69 to 83) to justify closing gap in China data (evidently restricted to 2002-2016) via statistical extrapolation, readers never see explanation for start year of 1963. Comes from US, via USGS? Or something from RoW? Readers can guess why data extend to 2020, but 1963 never explained and doesn't match other products.

**Response:** We are very grateful to the reviewer for this comment. In this study, in order to establish the lime carbon sink database, we collected data on the production of lime, paper and paperboard, crude steel, alumina in different countries and different years, the proportion of lime usage in various fields, such as chemical (including PCC production, sugar, paper, calcium carbide, others), metallurgy (crude steel, alumina, others), construction (lime stabilized soil, lime mortar, others), environmental protection and others in different countries and different years, the proportion data of paper white sludge, calcium carbide slag, steel slag, red mud utilization and CaO content, etc.

Considering that the lime production data in the United States can be traced back to 1930, and the lime production data for earlier years in China and the world are not recorded. Therefore, we decided to fill this gap by fitting methods, thereby extending the time scale of the study to 1930.

First of all, we fitted China's lime production. The only source of China's lime production statistics is the "China Building Materials Industry Yearbook", which records the lime production data from 1996 to 2020, of which the data from 2015 to 2018 is missing; in addition, the statistical yearbook also introduces the use of lime in various industries. From this, we know that the production of lime in construction, steel, calcium carbide and alumina in the downstream sector of lime accounts for more than 90% of lime production, which is the main factor affecting lime production. Therefore, we collected China's cement production (1930-2020), the completed area of housing in the whole society (1963-2020). 2020), steel production (1949-2020), calcium carbide production (1949-2020), and alumina production (1954-2020) were fitted to the lime production data. Taking China's lime production as the dependent variable, the stepwise linear regression method was used to construct a regression model. The independent variables entering the model include calcium carbide production, the completed area of houses in the whole society, cement production and alumina production. Since the completed area data of houses in the whole society was only extended to 1963, the model only predicted the lime production data from 1963 to 1995. Then, through the ARIMA (0,1,0) model, with external control variables including the steel production, calcium carbide production, and

cement production, we fit the lime production in China from 1949 to 1962 (the steel and calcium carbide production data were only extended to 1949). Finally, we use the ARIMA (2,2,0) model without external control variables to fit the lime production in China from 1930 to 1948. So far, we have obtained the fitted lime production data for China from 1930 to 2020.

After obtaining the Chinese lime production data, we corrected the global lime production data from the USGS from 1963 to 2020. The ARIMA (1,0,0) model was then used to fit the global lime production from 1930 to 1962 with global alumina production, steel production and cement production as external control variables.

As stated by the reviewer, the year 2020 as the termination year is related to the year of data update, and the starting year was set to 1963 for the following reason: In the report for cement from 1930 to 2019 in ESSD 2021 (Guo et al. https://doi.org/10.5194/essd-13-1791-2021), the data source for the cement production is United States Geological Survey (USGS), which provides data since 1930. Similarly, in our study, global and U.S. lime, crude steel, and alumina production data were obtained from the USGS: *Bureau of Mines Minerals Yearbook* (https://www.usgs.gov/centers/national-minerals-information-center/minerals-yearbook-metals-and-minerals). However, in this database, lime production data are recorded from 2000 for China and from 1963 for the world. To maintain the consistency of data years and maximize the time scale, we took 1963 as the starting year, and fitted the missing lime production data in China from 1963 to 2000 to build the lime carbon sink database for 1963 to 2020.

**Changes:** Line 69-93, Add 'The global lime production data came from Lime Statistics and Information (USGS, 2022), but the data did not include the statistics of China's lime production between 1963 and 1984. In addition, the statistical value of China's lime production from 1985 to 2001 was underestimated compared with the actual value (Cao et al., 2019), which led that the statistical data of global lime production during 1963-2001 was significantly less than the actual production (Fig. 2b). The lime production data of China in this study were obtained from (China Construction Material Industry Yearbook, 2022). Considering that lime production data is available for the United States since 1930, which is much earlier than the recorded data for China and the ROW, we filled gaps in the data using fitting methods, thereby extending the time scale of the study to 1930.

First, we fitted China's lime production. The only source of China's lime production statistics is the "China Building Materials Industry Yearbook", which records the lime production data from 1996 to 2020, of which the data from 2015 to 2018 is missing; in addition, the statistical yearbook introduces the use of lime in various industries. From this, we know that the production of lime in construction, steel, calcium carbide, and alumina in the downstream sector of lime accounts for more than 90% of lime production. Therefore, we collected data on China's cement production (1930–2020), the completed area of housing in the whole society (1963–2020), steel production (1949–2020), calcium carbide production (1949–2020), and alumina production (1954–2020) and fitted them to the lime production data. Taking China's lime production as the dependent variable, the stepwise linear regression method was used to construct a regression model. Since the completed area data of houses in the whole society was only available from 1963, the model predicted lime production data from 1963 to 1995. Then, through the ARIMA (0,1,0) model, with external control variables including the steel production, calcium carbide production, and cement production, we fit the lime production in China from 1949 to 1962 (the steel and calcium carbide production data were only extended to 1949). Finally, we used the ARIMA (2,2,0) model without external control

variables to fit the lime production in China from 1930 to 1948. From this, we obtained the fitted lime production data for China from 1930 to 2020 (Fig 2a).

After obtaining the Chinese lime production data, we corrected the global lime production data from the USGS from 1930 to 2020 (Fig 2b). The ARIMA (1,0,0) model was then used to fit the global lime production from 1930 to 1962 with global alumina production, steel production and cement production as external control variables.'

**2.Comments:** Problem with single years. Authors discuss 2020 lime emissions but with almost no discussion: a) they had to extrapolate - at least for China data - to get to that number; b) we know that any single year carries very large (larger than 95% CI) uncertainty; and c) we know that 2020 particularly, occurring as it did as countries emerged (or, not) from economic and social impacts of pandemic, itself represented an unusual year.

**Response:** Thanks very much for reviewer's opinion.

a) According to our calculation, the global $CO_2$ emissions of lime production process in 2020 are 293.99 Mt, of which China, the United States and other countries emit 211.00, 71.72 and 11.73 Mt, respectively. Numerically, compared with 2019, except the United States, the $CO_2$ emissions of China, the rest of word and the world total have slightly increased, but at a small growth rate.

b) As mentioned in our methodology, the annual $CO_2$ emissions of the lime industry are closely related to its production, and in the case of China, the lime production was not seriously affected by COVID-19 pandemic in 2020, and USGS statistics show the same production in 2020 as in 2019 (310 million tons, as shown in the table 1 below), but as reviewer said, the statistics have considerable uncertainty, where Table 2 shows the results of $CO_2$ emission distribution of China's lime production for 2018-2020 after 100,000 simulations, and it can be seen that although there is no difference between the production data of the two years, their carbon emission calculation results still have differences. The difference is about 0.02%. Considering the stochastic of the Monte Carlo simulation, this very small difference is acceptable.

Table 1 Lime production data for different regions 2018-2020

| Lime production | 2018 | 2019 | 2020 |
|---|---|---|---|
| The United States (thousand tons) | 18000 | 16900 | 15800 |
| China (thousand tons) | 300000 | 310000 | 310000 |
| The rest of world (thousand tons) | 106000 | 105100 | 101200 |
| World total (thousand tons) | 424000 | 432000 | 427000 |

Data from United States Geological Survey (USGS). Lime Statistics and Information. https://www.usgs.gov/centers/national-minerals-information-center/lime-statistics-and-information

Table 2 Distribution of China's lime production process $CO_2$ emission results from Monte Carlo 100,000 simulations for 2018-2020 (unit: Mt)

| percentile | 2018 | 2019 | 2020 |
|---|---|---|---|
| 2.5 | 180.63 | 186.48 | 186.64 |
| 5 | 183.74 | 189.88 | 189.85 |
| 50 | 204.08 | 210.97 | 211.00 |
| 95 | 225.15 | 232.65 | 232.66 |
| 97.5 | 228.70 | 236.44 | 236.37 |

c) 2020 is a remarkable year, according to the report of Global carbon budget 2021 (https://doi.org/10.5194/essd-14-1917-2022), Global fossil $CO_2$ emissions were 5.4 % lower in 2020 than in 2019, because of the COVID-19 pandemic, with a decline of 0.5 GtC to reach 9.5 ± 0.5 GtC (9.3 ± 0.5 GtC when including the cement carbonation sink) in 2020. Nevertheless, the carbon emission reductions in 2020 due to the COVID-19 are mainly in the transport sector, which is associated with the transport and logistics control measures to overcome the epidemic. However, lime industrial production had limited or no impact from the epidemic. From the production point of view, the COVID-19 pandemic does not affect the technological level, nor the supply capacity of lime industrial industry. In addition, the COVID-19 epidemic happened mainly during the gathering period of major festivals, such as the Spring Festival in China. This period is supposed to be the low season for industrial production, and enterprises will stop working, which further reduced the impact suffered by the lime industry. Therefore, the carbon emission during lime production in 2020 also showed a more stable trend.

**Changes:** Line 227-232, Add 'In general, the global lime output fluctuated and increased over time, from 139.62 Mt in 1930 to 394.93 Mt in 2019. In the early 1930s, the lime output decreased slightly, which may be due to the impact of the 'The Great Depression' and the closure of many factories, resulting in a decrease in the global lime output. In 2020, affected by the COVID-19, the lime production dropped slightly to 391.64 Mt (USGS: 427 Mt).'

Furthermore, we added pictures and descriptions related to lime production (Line 245).

[Figure]

**Figure 2: (a) Lime production in different countries or regions from 1930 to 2020. Shadows represent uncertainty ranges. CI (Confidence interval). (b) Global lime production from 1930 to 2020. (c) Annual $CO_2$ emissions from industrial processes.**

Line 278-279, Add 'Affected by the COVID-19, the amount of China's lime carbon sink decreased in 2020 compared with that in 2019 (about 24.94 Mt C).'

**3.Comments:** Impact estimate. Weaknesses in using specific single years notwithstanding, this reader evaluated 2021 GCB (https://doi.org/10.5194/essd-14-1917-2022). I find total 2020 emissions as 9.3 + 0.5 GtC/yr (using customary well-justified GCB + 5% uncertainty) when they include cement carbonation correction. Without carbonation correction they report 9.5 GtC/yr, so total cement carbonation term = 0.2 GtC/yr in 2020. Authors of this manuscript give 134.33 Mt CO2, converts to approximately 36 MtC per 2020. Given overall uncertainties of 500 MtC per year and total cement carbonation of 200 MtC per year in 2020, numbers presented here (36 / 200) represent at most 20% correction on carbonation (numbers provided by these authors in their Table 2 tend to support my estimates) and only 0.4% correction on overall emissions, well within uncertainty limits. Not clear to this reader why authors consider this work worthy of ESSD publication? Although they proclaim (lines 22-23) importance of lime processes in carbon cycle, actual numbers (which they never quote but should) prove otherwise.

On the other hand, we need to quantify all these corrections (however minor) so authors need to demonstrate special efforts and skills to certify extremely small numbers. Not clear to this reader how authors came up with 38% number (line 19) nor what "associated processes" (line 20) they refer to. Later they quote another author (line 42) that total lime-related carbonation occurs within only 34% of lime produced. Authors' obvious enthusiasm for their topic obscures practical impact; this reader had to resort to own calculation (above). Most readers will need a statement about minimal current impact, coupled with justification to quantify as precise albeit small correction and as possible future mitigation option.

**Response:** Thank you for your comments on this part. The global industrial process carbon emissions accounted for 24% of all direct anthropogenic emissions in 2019 (IPCC AR6), and the lime production process belongs to the industrial sector in the IPCC accounting. In addition to the carbon emissions caused by the burning of fossil fuels used for calcining limestone and purchased electricity (the fuel burning and electricity emissions are accounted in the energy sector), about 70% of carbon emissions in the lime industry are from the emissions of lime industrial processes, that is, the emitted from the production of lime by heating and decomposition of limestone $CaCO_3 \xrightarrow{heat} CaO + CO_2 \uparrow$). The 'associated process' (line 20) implied the lime process that generates carbon emissions, i.e. the lime production process. We agree with the reviewer's comment and in the next revision we will clarify the relevant statement from the reader's perspective.

According to Shan's study (https://doi.org/10.1016/j.apenergy.2015.04.091), lime production is the second largest industrial process carbon emission after cement production. Like cement production, lime will produce a large number of lime-based materials during its production, use and waste disposal. These materials are prone to carbonation under natural conditions. Therefore, when we study its carbon emissions, it is also very necessary to explore how large the carbon sink of this alkaline material is. Like cement, our calculation of carbon sequestration process of lime-based materials can compensate for the partial IPCC methodological deficiency of lime carbon sequestration. We compared the lime carbon sequestration with the lime industrial process carbon emissions. The lime carbon sequestration was 134.33Mt, accounting for 38.79% of the lime industrial process carbon emissions. Obviously, the lime carbon sink is of great significance to the lime industry and the industrial carbon balance. We agree with the reviewer, and will explicitly clarify the importance of the lime carbon sink for the carbon balance of the industrial sector using our own calculation in the next revision. Furthermore, through our estimation that the annual carbon sequestration of lime in 2020 is comparable to the global annual averaged carbon sequestration of siliceous rocks, it can be indicated that the carbon sink of lime-based materials had an incremental impact on the global carbon cycle year by year under the influence of human activities. This is also equivalent to approximately 1.03% of the global terrestrial annual carbon sink from 2010 to 2020, which can interpret the whereabouts of some global missing carbon sinks, and is of significance for the global carbon cycle research.

**Changes:** The clarification is adjusted as follows: A substantial amount of $CO_2$ is released into the atmosphere from the process of the high temperature decomposition of limestone to produce lime. However, during the life cycle of lime production, the alkaline components in lime will

continuously absorb $CO_2$ in the atmosphere when use and waste disposal. Here, we adopt an analytical model describing the carbonation process to obtain regional and global estimates of carbon uptake from 1930 to 2020 using lime lifecycle use-based material data. The results reveal that the global uptake of $CO_2$ by lime increased from 9.16 Mt C $yr^{-1}$ (95% confidence interval, CI: 1.84-18.76 Mt C) in 1930 to 35.27 Mt C $yr^{-1}$ (95% CI: 23.63-50.39 Mt C) in 2020. Cumulatively, approximately 1444.70 Mt C (95% CI:1016.24-1961.05 Mt C) were sequestered by lime produced between 1930 and 2020, corresponding to 38.83% of the process emissions during the same period, mainly contributed from the utilisation stage (76.21% of the total uptake). We also fitted the missing lime output data of China from 1930 to 2001, thus compensating for the lack of China's lime production (cumulative 7023.30 Mt) and underestimation of its carbon uptake (467.85 Mt C) in the international data. Since 1930, lime-based materials in China have accounted for the largest proportion (about 63.95%) of the global total. Our results provide data to support including lime carbon uptake into global carbon budgets and scientific proof for further research of the potential of lime-containing materials in carbon capture and storage. The data utilized in the present study can be accessed at https://doi.org/10.5281/zenodo.7628614 (Ma, 2022). (Line 11-24).

Line 329-332, 'Regarding the global carbon cycle, the annual carbon uptake by lime was approximately 1.65% of the average global forest ecosystem sink from 2001 to 2010, and can explain approximately 1.55% of the missing global carbon sink (2.37 Gt C $yr^{-1}$) (Global Carbon Budget 2021, 2022). Therefore, if the lime sink is incorporated, the global carbon budget, which already includes data for carbon sinks of the ocean, land, and cement can be improved.'

Amendment to the " how authors came up with 38% number (line 19) " issue. We updated the lime carbon sink data to 1930. We updated the lime carbon sink data to 1930, and 38% has been updated to 39.09%. Line 279-272, 'Cumulatively, 1444.70 Mt C (95% CI:1016.24–1961.05 Mt C) were sequestered by lime-containing materials between 1930 and 2020. This means that 38.83% of $CO_2$ emissions from the production process of calcining limestone process were offset by lime carbon uptake at the same stage (1930-2020).'

**4.Comments:** Please can I also suggest that: a) these authors need to reconsider how they can justify reporting unrealistic uncertainties at tenths or hundredth of Mt CO2); and b) work could have stronger impact if they adopted units (C rather than CO2) and uncertainty conventions of GCB.

**Response:** Thank you for your suggestions.

a) We used the Monte Carlo method to estimate the $CO_2$ uptake of lime materials and the ultimate result was measured by the statistical indicator MEDIAN, and the 95% confidence interval as a range of uncertainty. Therefore, the final result can only be considered as approaching the true value with a higher probability, but not the true value. This is the most probable approximation in the situation where the true value cannot be obtained.

b) It is correct that, as the reviewer suggested, that we can also use C to indicate the impact.

**Changes:** Line 209-217, 'We identified 16 groups of impact factors associated with the estimation of lime process carbon emission and carbon sequestration, which included 115 input-specific parameters, each with a specific statistical distribution (see the Supplementary Information). Due to the difficulty in obtaining the true values of the parameters, we employed the Monte Carlo approach recommended by the 2006 IPCC Guidelines for National Greenhouse Gas Inventories to access the uncertainties for the carbon emission and removal of lime materials. We fed the statistical characteristics of the 115 variables into our models, and the simulated carbon emission and removal

results were obtained through 10,000 iteration Monte Carlo simulation. Subsequently, statistical analysis was then performed to derive the median and the corresponding lower and upper bounds of the 95% confidence intervals (CI) for the carbon uptake and emission of lime materials.'

Furthermore, we have modified the unit of the whole paper from Mt $CO_2$ to Mt C.

**5.Comments:** ESSD references. It seems poor form to miss so much highly-relevant prior work, much of it cited above, published in the journal which these authors choose to use. Perhaps with focus on lime processes and lime engineering they have missed relevant climate impact literature? Fair enough if they want engineers as their audience but they clearly reach for climate relevance. If so, they need to cite relevant literature.

**Response:** thank you for your comments. At present, our paper mainly focuses on the carbon sink of lime under natural conditions, mainly from the perspective of data accounting. In addition, the research focus of our paper is also on the completion of lime deficiency data in China and the world. The relevant links to its mitigation of climate change are not the research focus of this paper.

**Changes:** We have added relevant references.

**6.Comments:** Posting uncertainties in separate file to actual data seems unusual at best? Derives from authors' mechanisms (after-the-fact Monte Carlo simulations) to estimate uncertainties? Validations very challenging for one-off products but I have given some budget-related hints above. Authors might know other validation options? Unfortunately, these authors never address validation. Perhaps they should review ESSD guidelines at https://www.earth-syst-sci-data. net/10/2275/2018/. I decided not to list further line-by-line technical changes. I vacillate between recommending rejection or recommending major revisions; I leave that question to handling editor. In reading this manuscript I wanted but could not find key justification and relevance nor acceptable uncertainties nor any validation attempt.

**Response:** In response to the questions about 'Posting uncertainties in separate file to actual data': We thank the reviewers for their careful and detailed comments, which prompted us to reconsider the presentation of the data from the reader's perspective. Putting the input parameters of the model, uncertainties, and simulation results in one file does seem to confuse the reader. In the next revision, we will make a change to post the uncertainties and Monte Carlo simulation results in separate files to more clearly show the sources of uncertainties, the processing and generation of the data, and the ultimate accounting results.

In response to the 'Questions validation': we thank the reviewer for the concern of data accuracy, and we fully agree with the importance of data accuracy. Our explanation is that in this paper, we employed the IPCC method, i.e., the inventory analysis method, to calculate the carbon sequestration of lime materials, i.e., the carbon sequestration of lime materials is equal to the activity level data multiplied by the sequestration factor, so it was not necessary to evaluate the accuracy of the model itself. The focus of our work was on how to minimize the range of uncertainties that exist in the parameters themselves, and improve the hit probability of the final results by reducing the parameters' uncertainties. As mentioned above, the reviewer presented a comparison of lime sinks with GHG emissions and with cement sinks; we also carried out a comparison of lime sinks with cement sinks and industrial process carbon emissions in our paper, but strictly speaking, these were not validations of the model. In the framework of Monte Carlo simulation, the modelled result is only the best possible approximation of the true value. it is a statistical probability, with stochastic characteristics,

therefore normally no validation is required.

**Changes:** None

**RC2**

**Summary and main impressions**: This study analyzes the global and regional uptake of $CO_2$ by lime through the different process of lime production from 1963 to 2020. This study is of interest for the global carbon community, as it is important to more accurately account for sources and sinks of $CO_2$ by lime-containing materials for better estimation of its impact on the carbon cycle. However, the manuscript is not clear for certain aspects of the study, a lot of information is missing, and more analyses should be done. I could not find the supplementary information, and the study mentions a model to conduct this work but no information on this model is provided in the manuscript. Please, find my comments below.

**Response:** Thank you for recognizing the importance of this piece of research. Regarding the two issues you mentioned, on the former, we have presented the detailed information, including activity level data and carbon absorption factor data, as well as uncertainty analysis results by the Monte Carlo simulations, in the dataset at https://doi.org/10.5281/zenodo.7628614. And we will further improve the data information and description in the next revision. On the latter, according to the diffusion characteristics of $CO_2$ in different types of lime materials, we used the method of greenhouse gas inventory analysis, combined with Fick's second law, to build the mathematical model of lime carbon sink. See formula 2-21 for details. Furthermore, we will address your concerns in the following texts.

**General comments**

**1.Comments:** Ln. 22. For the period 1963 to 2020, your results show that 38.79% of $CO_2$ emissions were sequestered by lime production from the total global $CO_2$ lime emissions. So, your results and this sentence suggest that global $CO_2$ sink from lime corresponds to ~40% with ~60% of global $CO_2$ source from lime. How, Ln 22-23, can you justify that lime materials is a global carbon sink that can reduce the carbon footprint of lime production? Need more clarification in the abstract.

Additionally, in the abstract you are only giving numbers of the $CO_2$ lime uptake at a global scale and for China. What about the $CO_2$ lime emissions for the 1963-2020 period for China and at global scale?

**Response:** Thank you for your suggestion. We readjusted the abstract,

**Changes:** Line11-24, 'A substantial amount of $CO_2$ is released into the atmosphere from the process of the high temperature decomposition of limestone to produce lime. However, during the life cycle of lime production, the alkaline components in lime will continuously absorb $CO_2$ in the atmosphere when use and waste disposal. Here, we adopt an analytical model describing the carbonation process to obtain regional and global estimates of carbon uptake from 1930 to 2020 using lime lifecycle use-based material data. The results reveal that the global uptake of $CO_2$ by lime increased from 9.16 Mt C yr$^{-1}$ (95% confidence interval, CI: 1.84-18.76 Mt C) in 1930 to 35.27 Mt C yr$^{-1}$ (95% CI: 23.63-50.39 Mt C) in 2020. Cumulatively, approximately 1444.70 Mt C (95% CI:1016.24-1961.05 Mt C) were sequestered by lime produced between 1930 and 2020, corresponding to 38.83% of the process emissions during the same period, mainly contributed from the utilisation stage (76.21% of the total uptake). We also fitted the missing lime output data of

China from 1930 to 2001, thus compensating for the lack of China's lime production (cumulative 7023.30 Mt) and underestimation of its carbon uptake (467.85 Mt C) in the international data. Since 1930, lime-based materials in China have accounted for the largest proportion (about 63.95%) of the global total. Our results provide data to support including lime carbon uptake into global carbon budgets and scientific proof for further research of the potential of lime-containing materials in carbon capture and storage. The data utilized in the present study can be accessed at https://doi.org/10.5281/zenodo.7628614 (Ma, 2023).'

**2.Comments:** Ln 65. You mentioned "this study significantly improves the global carbon uptake model" but what was improved and how? No information in the whole manuscript could be found on this improvement. More information should be provided on this improvement.

**Response:** We are grateful for the suggestion. To be more clearly and in accordance with the reviewer concerns, we have added a more detailed interpretation as following: Due to the lack of accounting methods and the difficulty in obtaining data, the lime carbon sink has not been revealed. According to our calculation, the annual carbon sequestration of lime accounts for about 1.03% of the global terrestrial carbon sequestration. In addition to cement and alkaline solid waste, lime carbon absorption is also a carbon sink generated by human activities. The research results can explain the whereabouts of some missing carbon sinks in the global carbon sink, and then make up for the global carbon sink model.

**Changes:** Line 217-227, The lime output of China and the world from 1930 to 2001 was re-fitted, and the deficiency of the international database was completed.

Line 330-335, 'In 2020, the global uptake of $CO_2$ by lime was equivalent to 2.15% of the global industrial emissions of $CO_2$; therefore, neglecting this sink caused an overestimation of the global carbon emission from industrial processes. Regarding the global carbon cycle, the annual carbon uptake by lime was approximately 1.65% of the average global forest ecosystem sink from 2001 to 2010, and can explain approximately 1.55% of the missing global carbon sink (2.37 Gt C $yr^{-1}$) (Global Carbon Budget 2021, 2022). Therefore, if the lime sink is incorporated, the global carbon budget, which already includes data for carbon sinks of the ocean, land, and cement can be improved.'

**3.Comments:** Ln.80. You assumed for the PCA calculation that you have a linear relationship between variables, but how would you justify that there is a linear relationship for the data over the period 1963-2000? Additionally, is the lime production linear over the period 1963-2020 for all three regions?

**Response:** Data of lime production in 2001-2016 is compiled by the Chinese Lime Association, the only institution that records the production, and their figures are based on the demand of downstream consumer sectors, including crude steel, construction and chemical industry. Among them, the amount of lime used in crude steel, calcium carbide, construction and alumina accounts for more than 90% of the total lime production. Therefore, we selected crude steel production, calcium carbide production, housing completion area and alumina production as the four variables for predicting lime production. Although there are no specific annual data records from 1963 to 2000, the use of lime in China has a long history. From the 7th century BC, lime became an important building material in China and began to be used, and according to the Chinese Statistical Yearbook, from 1963 to 2000, China was produced with calcium carbide, crude steel, alumina and other industrial products. Therefore, we assume that the data of China's lime production from 1963 to 2000 are still related to these four major industries.

Your question is very reasonable. We also used autoregressive models and other methods to predict the data from 1963 to 2000, but the method mentioned in this article obtained the largest coefficient of determination.

For the other three regions, we have carried out the same analysis for the United States, the world and other countries. According to USGS records, the downstream use sectors of lime in the United States mainly include construction, chemical industry, metallurgy, environmental protection and refractory materials. We collected the crude steel production, alumina production and cement production in the United States from 1963 to 2020. The correlation between each variable is shown in the table below. It can be seen that there is a strong correlation between the lime production in the United States and each variable. In the construction of a general linear model, the determining coefficient of the model is 0.636.

**Table 1** Correlation of various variables in the United States from 1963 to 2020

|  | crude steel production | alumina production | cement production | lime production |
|---|---|---|---|---|
| crude steel production | 1 |  |  |  |
| alumina production | 0.827** | 1 |  |  |
| cement production | -0.017 | -0.036 | 1 |  |
| lime production | 0.335* | 0.289* | 0.716** | 1 |

**. Correlation is significant at the 0.01 level (2-tailed).

*. Correlation is significant at the 0.05 level (2-tailed).

Likewise, for other countries, there is a strong positive correlation between lime yield and other variables. A general linear model is constructed with lime yield as the dependent variable and other variables as the independent variable, and its determination coefficient is 0.749.

**Table 2** Correlation of variables in ROW

|  | crude steel production | alumina production | cement production | lime production |
|---|---|---|---|---|
| crude steel production | 1 |  |  |  |
| alumina production | 0.942** | 1 |  |  |
| cement production | 0.903** | 0.951** | 1 |  |
| lime production | 0.782** | 0.613** | 0.602** | 1 |

** Correlation is significant at the 0.01 level (2-tailed).

From 1963 to 2020, the global lime production has a strong positive correlation with the global crude steel, alumina, and cement production. Taking the global lime production as the dependent variable and other variables as independent variables, the obtained model has strong collinearity. Therefore, the principal component analysis method is selected to obtain a principal component, and the coefficient of determination obtained by the model is 0.946.

**Table 3** Correlation matrix between global lime production and various variables

| | crude steel production | alumina production | cement production | lime production |
|---|---|---|---|---|
| crude steel production | 1 | | | |
| alumina production | 0.984** | 1 | | |
| cement production | 0.985** | 0.986** | 1 | |
| lime production | 0.980** | 0.964** | 0.959** | 1 |

** Correlation is significant at the 0.01 level (2-tailed).

Therefore, this linear relationship is also very obvious on a global scale, but this linear relationship is relatively weak in the United States and other countries.

**Changes:** None

**4.Comments:** Ln. 96. "Considering the availability of lime production data", what do you specifically mean by availability of the data? Additionally, what are the uncertainties associated with the dataset used in this study? Any uncertainty in the lime production data of China and US?

**Response:** According to the USGS, the data of global lime production began in 1963. And the " availability of lime production data " mentioned in this paper refers to the data currently available to statistical institutions or literature journals. The uncertain detailed data of the lime production data of China and US, which are included in the supplementary materials. The data utilized in the present study can be accessed at https://doi.org/10.5281/zenodo.7112485 (Ma, 2022)

**Changes:** None

**5.Comments:** Ln 96 and Equation 1. Why are you using method 1 and not method 2 (which use a correction factor) or 3 here? Should be mentioned in the text. The indexes "l" and "i" in your equation are not defined.

**Response:** We are very grateful to the reviewer for this comment. As we mentioned above, Tier 1 is based on the default emission factor of the national lime production process provided by the IPCC, and the proportion of usage of various lime uses; the emission factor in tier 2 is collected by national or plant level data; for Tier 3, the emission factor should be specific to the carbonate consumption process in each factory. The lime production data in this study belong to the national level, and the emission factor uses the default emission factor provided by the IPCC, which is not detailed to the factory level of each country, so Tier 1 method is selected; Thanks very much for reviewer's opinion. In Equation 1, l refers to different types of lime use, including PCC, sugar making, lime-stabilized soil and lime mortar, and i refers to different years. We will add it in the revision.

**Changes:** Line 104-105, 'Lime comes from the decomposition of limestone in shaft or rotary kiln, and the carbon emission of this industrial process can be estimated from using the IPCC method (IPCC, 2006).'

Line 109-110, add 'l refers to different types of lime use, including PCC, sugar making, lime-stabilized soil and lime mortar, and i refers to different years.'

**6.Comments:** Ln 101. Is this method for estimating emissions factors based on tier1 method

from IPCC 2006? If yes, which tier method for choosing the emissions factor are you using exactly?

Emission factor for combustion have values depending on the category and on the target year. Are the values of emission factor used for lime production and in this line for specific year and specific process type?

Based on the table available from IPCC 2006 (https://www.ipcc-nggip.iges.or.jp/EFDB/find_ef.php?ipcc_code=2.A.2&ipcc_level=2), developing countries emission factor for dolomitic lime production is equal to 0.77 t $CO_2$/t lime while developed countries value is of 0.86. The emission factors mentioned in your paper do not correspond to those from IPCC database, can you develop why?

What are the uncertainties associated with the emission factor and how did you count of these uncertainties in your study?

**Response:** I'm very sorry for the careless typo here. The emission factors used in our actual calculation are 0.683 for China, 0.77 for the United States, and 0.75 for other countries. Among them, we use IPCC2006 tier1 for the emission factors of the United States and other countries, and China's emission factor is based on China's provincial greenhouse gas emission guidelines, which is consistent with the research of Shan and is more in line with China's reality.

In the IPCC guidelines, Equation 2.8 illustrates how to calculate the Tier 1 emission factor for lime production.

$$EQUATION 2.8$$
$$EF_{lime} = 0.85 \times EF_{high\ calcium\ lime} + 0.15 \times EF_{dolomittic\ lime}$$

the EF $_{high\ calcium\ lime}$ is 0.75, and the default EF for dolomitic may be 0.86 or 0.77 for developed countries and developing ones. Substituting it into the formula, the emission factors of developed and developing countries are obtained as 0.77 and 0.75, respectively.

When setting the uncertainty of emission factors for different countries, we considered the emission factors for all lime types mentioned in the IPCC guidelines. The minimum value of the emission factor is 0.59, that is, the hydraulic lime type, and the maximum emission factor is 0.86 for developed countries. For developing countries, it is 0.77. Therefore, in our model, the maximum and minimum emission factors for China and ROW are 0.77 and 0.59, respectively, and 0.86 and 0.59 for the United States.

Changes: Line 111-115, 'Emission factors for the lime industry processes were determined using the composition of raw materials and the production technology. In the present study, 0.77-, 0.683-, and 0.75-ton $CO_2$ per ton of lime produced were adopted as emission factors for the US, China, and ROW, respectively (IPCC, 2006). Emission factors for the U.S. and ROW were according to the IPCC guidelines, whereas that for China was from the National Development and Reform Commission of China (Guidelines for provincial greenhouse gas inventories, 2011).'

**7.Comments:** Equation 3. According to IPCC 2006, the amount of dust depends on the type of kiln used in lime production. Have you considered the type of kiln in the parameter r$_{lkd}$ of your equation?

**Response**: Equation 3. The ash yield of different lime kilns is mainly obtained through literature collection (the ash yield of lime kilns is between 0.09 and 0.1). As an important calculation parameter of lime kiln ash, it participates in the simulation of 10000 times by Monte Carlo method.

**Changes:** None

**8.Comments:** Section 2.3. In this section, there is no information on where all parameters used for the equation come from. For instance, there is no information on the rate of CaO to $CaCO_3$ in dust. More information on these parameters should be inserted.

**Response**: Thank you for your suggestion. Because there are many parameters involved in this step, we put the relevant reference sources in the SI file, and the link is https://doi.org/10.5281/zenodo.7628614. We are happy to add this part in the next revision.

Ln193. We can agree on this. The lime carbon sink model here is constructed according to the diffusion law of $CO_2$ in different lime materials. For the carbon sink model of each material, see the public trial 2-21. The Ln 193 part represents the accounting method of the annual lime carbon sink. The description of the carbon sink will also be added in the next revision.

**Changes:** Line 355-357, add 'All the original datasets of $CO_2$ uptake by lime are available at https://doi.org/10.5281/zenodo.7628614 (Ma et al.,2023). This dataset contains three data files, including lime material production and uses, lime carbon emission and uptake results, and the uncertainty of lime carbon emission and uptake.'

**9.Comments:** Ln. 193. "[…] were utilized as inputs for the model (see the Supplementary Information)." First, I could not find the Supplementary Information. Second, there was no information of the model used for this study so far. Model name, model goal, model algorithm, … were not introduced so far. There should be a section to describe the model used for this study.

**Response**: We can agree on this. The lime carbon sink model here is constructed according to the diffusion law of $CO_2$ in different lime materials. For the carbon sink model of each material, see the equation 2-21. The Ln 193 part represents the accounting method of the annual lime carbon sink. The description of the carbon sink will also be added in the next revision.

**Changes:** Supplementary Information can be available at https://doi.org/10.5281/zenodo.7628614. The carbon sink accounting model is described in detail in lines 119-207.

**10.Comments:** Ln. 195. This expression is not clear. Why use the year ti-1 to calculate the uptake of $CO_2$ in year ti? It is not clear also how the contribution of annual uptake of carbon to the total carbonation can be calculated by using this expression. Please, clarify.

**Response:** Because different lime materials have different carbon absorption rates, some alkaline solid wastes with large particle size cannot be completely carbonized within one year. Therefore, to calculate the carbon sink of lime materials in the current year, the accumulated carbon sink of lime in $t_i$ year should be subtracted from the carbon sink of lime in $t_{i-1}$ year.

**Changes:** We added a description of this part in Line 204-208 and Line in 346-353.

**11.Comments:** Section 2.5. The uncertainty analysis is not enough detailed. The carbon absorption factors, and activity level data are only introduced here. More information on these parameters is needed. More information on the model uncertainty and statistics should be found here and not in supplement information (which cannot be found).

As mentioned in IPCC 2006, complete activity data are needed. Omission of use or lime production as a non-marketed intermediate, not well accounted for in inventories, may lead to an underestimation of lime production by country by a factor of 2 or more. Uncertainties associated with LKF are also non negligible. Have you accounted for these uncertainties by country?

A section should give information on the carbon sequestration analytical model used for this

work.

**Response:** Uncertainty analysis description and carbon sequestration analytical model have been improved in the new manuscript.

**Changes:** Line 209-216, add 'We identified 16 groups of impact factors associated with the estimation of lime process carbon emission and carbon sequestration, which included 115 input-specific parameters, each with a specific statistical distribution (see the Supplementary Information). Due to the difficulty in obtaining the true values of the parameters, we employed the Monte Carlo approach recommended by the 2006 IPCC Guidelines for National Greenhouse Gas Inventories to access the uncertainties for the carbon emission and removal of lime materials. We fed the statistical characteristics of the 115 variables into our models, and the simulated carbon emission and removal results were obtained through 10,000 iteration Monte Carlo simulation. Subsequently, statistical analysis was then performed to derive the median and the corresponding lower and upper bounds of the 95% confidence intervals (CI) for the carbon uptake and emission of lime materials.'

**12.Comments:** Ln.207-211. You mentioned a decrease in emissions in 2009 due to the financial crisis of 2008. But we can also observe a decrease in 2017, 2018, and some other years before. What are the reasons for all these decreased emissions?

Additionally, what is the reason for the sudden increase in emissions starting 2002? Ln. 213 seems to answer this question, but it is not clear if the increase mentioned Ln. 213 refers to the large increase starting in 2002.

You should give a description of the net emissions here. As we can see, net emissions (process + uptake) show net source of $CO_2$ emission at global scale.

It is not clear also if these results are based on the inventories or your model results. Please, clarify.

"Subtraction of the amount of $CO_2$ absorbed from $CO_2$ emissions", it is not clear which $CO_2$ absorbed from $CO_2$ emissions are you talking about. Please, clarify.

**Response:** (1)Reasons for the decline in other years: In addition to 2008, other significant carbon emissions decline years include 1982, 1990, 1993, 2002 and 2016.

(2)China's lime carbon emissions have risen sharply since 2002, with an average annual growth rate of 4.86% from 2002 to 2020, compared with an average annual growth rate of 0.65% from 1963 to 2002. This is mainly due to the steady growth of China's macro economy after 2002.

(3)The results in Ln 204-211 of the article are based on our model's calculations, which we will describe in a revised draft thanks to the expert's reminder.

(4)Thank the experts for their suggestions on net emissions, and we will add this part to the revised version.

**Changes:** Line 228-260, add 'Figure 2c shows the estimated $CO_2$ emissions from lime production processes in China, the U.S., ROW and globally from 1930 to 2020. According to our calculations, the global process $CO_2$ emissions increased from with 27.39 Mt C yr$^{-1}$ (95% Confident Interval, CI: 8.87–46.86 Mt C) in 1930 to 75.73 Mt C yr$^{-1}$ (95% CI: 69.18–82.33 Mt C) in 2020. In the early 1930s, carbon emissions slightly declined due to the impact of lime production and its uncertainty. The uncertainty of lime output can be transferred to the final simulation results of lime carbon emissions. The greater uncertainty of the parameters will lead to greater uncertainty in the simulation results. The results of the 10,000 iteration Monte Carlo simulation show the change trend (Figure 2c). On a global scale, emissions doubled from 44.63 Mt C yr$^{-1}$ in 2002 to 75.73 Mt C yr$^{-1}$

in 2020. During this period (2002–2018), the average annual rate of increase was 2.98%, which was significantly higher than the rate for 1930–2002 (0.68%). The cumulative emissions of $CO_2$ from 1930 to 2020 were 3720.16 Mt C (95% CI: 3166.18–4287.43 Mt C). Emissions decreased in 2009, which was likely caused by the global financial crisis in 2008, during which downstream lime industries experienced severe problems, such as excess produce, low production quantities, and stiff competition (Dong et al., 2010).

[Figure]

**Figure 2: (a) Lime production in different countries or regions from 1930 to 2020. Shadows represent uncertainty ranges. CI : Confidence interval. (b) Global lime production from 1930 to 2020. (c) Annual $CO_2$ emissions from industrial processes.**

$CO_2$ emissions from 1930 to 2020 in China account for approximately half of the global total. China was primarily responsible for the increase in the global emission from lime production processes during the studied period. In China, from 1930 to 2020, the average annual lime process $CO_2$ emission was 23.08 Mt C yr$^{-1}$, with 1.06% average annual growth rate. Notably, a rapid global increase in $CO_2$ emissions started in 2002. From 2002 to 2020, the average annual growth rate of carbon emissions from lime was 4.03%, which was far higher than that of 1930 to 2001 (0.32%). This was mainly due to the steady growth of China's macro economy after 2002. This finding was consistent with estimates from studies on the uptake of carbon by cement carbon based on similar approaches (Cui et al., 2019). These results are closely linked to the development of downstream sectors of the lime industry in China, such as the iron and steel, light and chemical, construction and materials industries (Shan et al., 2016b). In 2020, $CO_2$ emissions in China from lime production processes reached 49.93 Mt C yr$^{-1}$ (95% CI: 44.18–55.94 Mt C), and the cumulative emission was 2100.39 Mt C (95% CI: 1606.96–2620.93 Mt), accounting for 56.33% of the global total. This figure is higher than the 46.91 Mt C yr$^{-1}$ forecasted for 2020 by Tong et al. (2019), which can be attributed mainly to the emission reduction scenarios considered.

In the U.S., from 1930 to 2020, $CO_2$ emissions from lime production processes remained at around 2.72 Mt C yr$^{-1}$, and the cumulative emissions by 2020 were approximately 247.30 Mt C, which represents 6.63% of the global total. This relatively low value is because of a fairly stable production of lime in the U.S. and significant import of lime from Canada (USGS, 2022). Relatedly, for the ROW, the cumulative emission was 1380.77 Mt C, which represents 37.03% of the global total.'

**13.Comments:** Ln 214. There should be information as well about the period 2002-2020 for comparison with the global annual average.

You considered ROW as regions of developing countries Ln. 101. However, countries considered as developed countries might be included in it, such as Canada. Ln. 225, you mentioned there is "significant

import of lime from Canada" to the US. Is Canada part of ROW region? If yes, can you really

consider ROW as developing countries?

If ROW region was grouped into two categories (developed and developing countries), what would be the $CO_2$ emissions for both categories?

**Response:** Thanks to the advice of reviewer, the average annual growth rate of China's lime carbon emissions from 2002 to 2020 was 4.03%, which was higher than the global average annual growth rate of 2.98% during the same period.

Other countries are mainly other regions except China and the United States, without carefully distinguishing between developed and developing countries

**Changes:** Line 238-240, 'From 2002 to 2020, the average annual growth rate of carbon emissions from lime was 4.03%, which was far higher than that of 1930 to 2001(0.32%). This was mainly due to the steady growth of China's macro economy after 2002.'

Line 246-248, add 'From 2002 to 2020, the average annual growth rate of carbon emissions from lime was 4.03%, which was far higher than that of 1930 to 2001(0.32%). This was mainly due to the steady growth of China's macro economy after 2002.'

**14.Comments:** Ln. 231. What is the percentage of uptake of $CO_2$ compared to the emissions of $CO_2$ at global and regional scales?

Figure 3.b. Results from this figure have not been used. What is this figure used for and telling us?

**Response:** Cumulatively, 1444.70 Mt C (95% CI:1016.24–1961.05 Mt C) were sequestered by lime-containing materials between 1930 and 2020. This means that 38.83% of $CO_2$ emissions from the production process of calcining limestone process were offset by lime carbon uptake at the same stage (1930-2020). The highest sequestration was in China (~63.95%, 918.41 Mt C) because of the associated high production of lime materials (China Statistical Yearbook, 2022), followed by the ROW (~34.35%, 474.35 Mt C) and US (~3.01%, 43.28 Mt C). As for different regions, the percentage of uptake of $CO_2$ compared to the emissions of $CO_2$ :56.33% in China, 6.63% in U.S., and 37.03% in the rest of world.

We also added the description of Figure 3b

**Changes:** The percentage of uptake of $CO_2$ compared to the emissions is illustrated in Line 266-279. 'Figure 3b shows the annual uptake of $CO_2$ in different regions, where the area represents the cumulative uptake in each region under natural conditions. In the early 1930s, the carbon sink of lime was affected by the uncertainty of lime production parameters, and the trend was slightly decreased, which was similar to the change of carbon emissions in lime industrial process.'

Line 296-300, add 'Among the stages of the lime cycle, the service stage accounted for the highest uptake of $CO_2$ (1076.97 Mt C) from 1930 to 2020, representing 76.21% of the total. The uptake of $CO_2$ during the production and waste disposal stages were 36.95 and 299.19 Mt C, respectively (Fig. 5).

[Figure]

**Figure 5: Annual uptake of carbon dioxide by lime in different stages of its cycle.'**

**15.Comments:** Ln. 247. Should be Fig.3b

**Response: Thank you for your suggestion**

**Changes:** Figure 3.b. mainly shows the carbon sequestration of lime in different life cycle stages. We have revised 'Figure 2.b' to 'Figure 5' in Line 300.

**16.Comments:** Ln. 248. This sentence is not clear. Do you mean, the uptake of $CO_2$ by dust decreases or increases since 1963? Uptake of $CO_2$ by dust in 2020 is available but value for 1963 should be mentioned here as well.

**Response:** Well, the carbon absorption of dust is increasing. We will add the carbon absorption in 1930 to further improve the description.

**Changes:** Line 301-305, The changed content is 'Since 1930, the production stage is associated with a significant output of lime kiln dust (LKD), which is a by-product of the production of lime. The uptake of $CO_2$ by LKD in 2020 was 0.74 Mt C. This contribution is attributed to the development of the lime industry and increase in the disposal of LKD in landfills (Latif et al., 2015). The concentration of CaO in the ash of lime kilns is approximately 54.88%, and thus, this continuously absorbs $CO_2$ in landfills (Bobicki et al., 2012).'

**17.Comments:** Ln. 270. According to your results, "the uptake of lime-containing materials rapidly increased from 1963 to 2020", but you mentioned earlier as well that this increase is proportional to the lime-containing materials uses and production. Does the uptake increase proportional to the emission during the period, or is this uptake increasing at a certain point? Maybe your results should be displayed by removing the trends, or additional detrending results should show the uptake and emissions of $CO_2$ during all stages of the lime cycle, by region and a global scale. With your current results, it is difficult to say if there is or is not an increase in $CO_2$ uptake by lime-containing materials.

**Response:** Lime carbon sink and industrial process emissions show an upward trend from the trend. We show the change trend of emissions and absorption in different regions in Figure 3cnd Figure 4b. The carbon emission of lime refers to the emission of limestone calcination production, which is the carbon emission of industrial process. In Figure 5 we show the carbon absorption in the global life cycle. The carbon sink results calculated by us increase with time, which can explain this situation.

**Changes:** None

**18.Comments:** Ln. 275. You mentioned that if the lime sink would be incorporated in the global carbon budget (GCB), the carbon sinks could be improved, however, let's not forget that the

emissions of $CO_2$ associated with the calcination of lime and limestone outside of cement production is not considered in the GCB. Knowing that the net emissions from lime production are a source of $CO_2$, if the lime source and sink were included in the GCB, how much do you think the net emission from the GCB would change? Discussion on this aspect should be included here.

**Response:** Thank you for your suggestion. We discussed this part from the perspective of the proportion of lime carbon sink to global terrestrial carbon sink.

**Changes:** Line 332-334, add 'Regarding the global carbon cycle, the annual carbon uptake by lime was approximately 1.65% of the average global forest ecosystem sink from 2001 to 2010, and can explain approximately 1.55% of the missing global carbon sink (2.37 Gt C $yr^{-1}$) (Global Carbon Budget 2021, 2022). Therefore, if the lime sink is incorporated, the global carbon budget, which already includes data for carbon sinks of the ocean, land, and cement can be improved.'

**19.Comments**: Ln. 285 and Figure 5. Additional results should show the cumulative sources of $CO_2$ as well as the cumulative net $CO_2$ emissions. And further discussion should be added about the net emissions.

**Response:** The cumulative sources of $CO_2$ as well as the cumulative net $CO_2$ emissions will be showed in the next revised draft. And we will add further discussion about the net emissions.

**Changes:** The expression of this part is changed to 'Cumulatively, 1444.70 Mt C (95% CI:1016.24–1961.05 Mt C) were sequestered by lime-containing materials between 1930 and 2020. This means that 38.83% of $CO_2$ emissions from the production process of calcining limestone process were offset by lime carbon uptake at the same stage (1930-2020).'

**20.Comments:** Ln. 300. How can we consider lime as a carbon sink when the net emissions show a source of $CO_2$?

**Response:** In fact, our logic is: In our study lime carbon sink has been revealed, which is one of the important destinations of carbon emissions in the lime industry process, in other words, some industrial process carbon emissions are absorbed back by lime. If carbon sink is considered in the carbon emission accounting of industrial process, we can determine the net carbon emission, then we have solved the problem that the lime industrial process is overestimated.

**Changes:** None

> **Specific Comments:**

**1.Comments:** Ln. 14. Which existing data are you referring to? You are analyzing these data (satellite, in situ?) in your study, so you should mention here the data used (names of data?).
This sentence contains twice materials ("materials associated with the production […] of lime-containing materials"), could you rephrase?

**Response:** The existing data refer to the data available in the national statistical yearbook, literature journals, international databases, etc. This part of data is included in the annex, and we will display it in the text according to your suggestions.

**Changes:** We restated the introduction of data in lines 358-360.And the sentence of abstract has been modified as 'Here, we adopt an analytical model describing the carbonation process to obtain regional and global estimates of carbon uptake from 1930 to 2020 using lime lifecycle use-based material data. ' (line 13-14)

Ln. 15. The model used should be mentioned here.

**Response:** Thank you for your suggestion.

**Changes:** Line 13, add 'an analytical model describing the carbonation process'

**2.Comments:** Ln 20. Which associated process are you mentioning? Is this process the production of lime materials?

**Response:** Yes. This process refers to the industrial process of lime carbon emission.

**Changes:** We reintroduced it in the line 11, 'A substantial amount of $CO_2$ is released into the atmosphere from the process of the high temperature decomposition of limestone to produce lime.'

**3.Comments:** Ln. "Total uptake", do you mean total global uptake?

**Response:** Yes. Total global uptake.

**Changes: None**

**4.Comments:** Ln 62. The three stages should be mentioned here for clarification:

Limestone calcination (lime production)

Hydration reaction (lime decomposition)

Lime carbonation

**Response:** In fact, the three stages of this study are mainly the carbonization of lime in the three stages of production, use and waste disposal. Thank you for your comments. We have also re-described these three stages in the revised version.

**Changes:** Line 61, 'during the three stages (production, use and waste disposal process) of the lime cycle from 1930 to 2020.'

Line 296-299, 'Among the stages of the lime cycle, the service stage accounted for the highest uptake of $CO_2$ (1076.97 Mt C) from 1930 to 2020, representing 76.21% of the total. The uptake of $CO_2$ during the production and waste disposal stages were 36.95 and 299.19 Mt C, respectively (Fig. 5).'

**5.Comments:** Ln. 72. "[…] data on lime production from 1963 to 2000 in China were not available in the existing databases." What do you mean? Regarding the following sentence, you should mention that there was no data during this period in the China Statistical Yearbook.

**Response:** Yes, your statement is more precise. We have revised the year and related expressions in the article, extending the starting year to 1930.

**Changes:** Line 69-76, 'The global lime production data came from Lime Statistics and Information (USGS, 2022),  but the data did not include the statistics of China's lime production between 1963 and 1984. In addition, the statistical value of China's lime production from 1985 to 2001 was underestimated compared with the actual value (Cao et al., 2019), which led that the statistical data of global lime production during 1963-2001 was significantly less than the actual production (Fig. 2b). The lime production data of China in this study were obtained from (China Construction Material Industry Yearbook, 2022). Considering that lime production data is available for the United States since 1930, which is much earlier than the recorded data for China and the ROW, we filled gaps in the data using fitting methods, thereby extending the time scale of the study to 1930.'

**6.Comments:** Ln. 79. What are xxx and xx? Need more precision. "A linear regression was then built with xx", what does xx correspond to?

**Response:** We have revised the estimation of lime production simulation. We have adopted the multivariate linear regression method. In order to avoid multicollinearity, a stepwise method is used to establish a model.

**Changes:** Line 77-92, add 'First, we fitted China's lime production. The only source of China's lime production statistics is the "China Building Materials Industry Yearbook", which records the

lime production data from 1996 to 2020, of which the data from 2015 to 2018 is missing; in addition, the statistical yearbook introduces the use of lime in various industries. From this, we know that the production of lime in construction, steel, calcium carbide, and alumina in the downstream sector of lime accounts for more than 90% of lime production. Therefore, we collected data on China's cement production (1930–2020), the completed area of housing in the whole society (1963–2020), steel production (1949–2020), calcium carbide production (1949–2020), and alumina production (1954–2020) and fitted them to the lime production data. Taking China's lime production as the dependent variable, the stepwise linear regression method was used to construct a regression model. Since the completed area data of houses in the whole society was only available from 1963, the model predicted lime production data from 1963 to 1995. Then, through the ARIMA (0,1,0) model, with external control variables including the steel production, calcium carbide production, and cement production, we fit the lime production in China from 1949 to 1962 (the steel and calcium carbide production data were only extended to 1949). Finally, we used the ARIMA (2,2,0) model without external control variables to fit the lime production in China from 1930 to 1948. From this, we obtained the fitted lime production data for China from 1930 to 2020 (Fig 2a).

After obtaining the Chinese lime production data, we corrected the global lime production data from the USGS from 1930 to 2020 (Fig 2b). The ARIMA (1,0,0) model was then used to fit the global lime production from 1930 to 1962 with global alumina production, steel production and cement production as external control variables.'

**7.Comments:** Ln. 94-96. The sentence needs to be rephrased.

**Changes:** Rewritten sentence: Lime comes from the decomposition of limestone in shaft kiln or rotary kiln, and the carbon emission of this industrial process is estimated from the IPCC method (IPCC, 2006).

**8.Comments:** Ln.98. "in the present", what do you mean by it? Today?

**Response:** It refers to the current stage, or today.

**Changes:** None.

**9.Comments:** Ln. 105. A reference is missing here.

**Response:** Thanks to the advice of reviewer, the references here is "PR China National Development and Reform Commission. Guideline for provincial greenhouse gas inventories;

**Changes:** Line 115, add '(Guidelines for provincial greenhouse gas inventories, 2011).'

**10.Comments:** Equation 2. what is "l' referring to?

**Response:** Equation 2. 'l' in Equation 2. refers to lime

**Changes:** Line 109-110, add 'l refers to different types of lime use, including PCC, sugar making, lime-stabilized soil and lime mortar, and i refers to different years.'

**11.Comments:** Ln. 117. Units are missing

**Response:** The problem has been corrected. Unit: g/mol

**Changes:** Line 129, '56 and 12 g/mol, respectively'

**12.Comments:** Ln. 155. How many years are sufficient consequently?

**Response:** It usually takes several years or even decades. The duration is reflected in the time lag effect of lime carbon sink.

**Changes:** Line 345-346, 'Lime materials, such as MOR and SS, similar to cement and natural materials, also serve for the removal of $CO_2$ from the atmosphere for several years or decades (Fig. 6)'

[Figure]

**Figure 6: Worldwide annual uptake of atmospheric CO₂ by lime, disaggregated by years of production**

**13.Comments:** Equation 11. Indexes "m" and "p" are not defined.

**Response:** Thank you for your reminding.

**Changes:** Line 172-174, 'where $\varepsilon_{m,ij}$ denotes the mass of CaO in wastes (m,j=lime mud or red mud) that can be carbonated in year i, $m_{p,ij}$ is the quantity of paper and paperboard/alumina that are produced in the i-th year, and p is the production. $r_{m,ij}$ is the output rate of waste j and $f_{m,j}^{CaO}$ represents the concentration of CaO in waste j.'

**14.Comments:** Ln. 182. Only "a" and "b" are in equation 18. Where are "$a$" and "$b$"?

**Changes:** Line 194, a and $b$ represent the corresponding minimum and maximum diameters of SS particles in a given size distribution.

**15.Comments:** Ln. 193. "were utilised" should be "were utilized"

**Response:** Thank you for your reminding.

**Changes:** Change 'utilised' to 'utilized'

**16.Comments:** Figure 1 was not used in the manuscript.

**Response:** Figure 1. mainly shows the situation of different life cycle stages of lime and application of lime in different fields

**Changes:** Line 34, 'The enormous quantity of lime produced in the world (~427 Mt in 2020; USGS, 2022) is mainly employed in the following sectors (Figure 1): (1) chemical industry, such as for the production of precipitated calcium carbonate (PCC), manufacturing of paper, and refining of sugar; (2) environmental remediation/treatment, including water treatment, acid mine drainage, and flue gas desulphurization; (3) metallurgical industry, for instance as a fluxing agent in the production of iron and steel; and (4) construction industry for building materials including lime mortar and lime-stabilized soil-asphalt mixtures (National Lime Association, 2020).'

**17.Comments:** Ln. 204. "and at a global scale" should also be mentioned.

**Response:** Yes.

**Changes:** Line 232, add 'Figure 2c shows the estimated CO₂ emissions from lime production processes in China, the U.S., ROW and at a global scale from 1930 to 2020.'

**18.Comments:** Ln. 445. (a) should mention "at a global scale"

**Response:** Thank you for your suggestion. We have readjusted the picture content

**Changes:** 'Figure 2: (a) Lime production in different countries or regions from 1930 to 2020. Shadows represent uncertainty ranges. CI: Confidence interval. (b) Global lime production from 1930 to 2020. (c) Annual $CO_2$ emissions from industrial processes.'

**19.Comments:** The dataset used for US, China and ROW emissions were defined in the methodology but not the global annual emissions.

**Response:** Thank you for your comments. We have added a description of global carbon emissions

**Changes:** Line 233-237, add 'According to our calculations, the global process $CO_2$ emissions increased from with 27.39 Mt C yr$^{-1}$ (95% Confident Interval, CI: 8.87–46.86 Mt C) in 1930 to 75.73 Mt C yr$^{-1}$ (95% CI: 69.18–82.33 Mt C) in 2020. In the early 1930s, carbon emissions slightly declined due to the impact of lime production and its uncertainty. The uncertainty of lime output can be transferred to the final simulation results of lime carbon emissions. The greater uncertainty of the parameters will lead to greater uncertainty in the simulation results.'

**20.Comments:** Figure 2.a. What are the shadows representing? Information on this shadow should be included in the caption of the figure.

**Response:** Shadows represent uncertainty ranges.

**Changes:** Line 245 and 286, add 'Shadows represent uncertainty ranges.'

**21.Comments:** Ln. 232. "whereas the area represents the cumulative uptake in each region under natural conditions." it is not clear which area you are talking about. Which natural conditions are you mentioning? For which process?

**Response:** We have made major modifications to the results and elaborated on the contents you mentioned. The natural conditions here mainly refer to the atmospheric exposure environment.

**Changes:** Relevant modifications are shown in Section 3.2

**22.Comments:** Ln. 205. The meaning of CI (Confidence Interval) should be mentioned in the text.

**Response:** Thank you for your suggestion. We have explained this part in the uncertainty analysis.

**Changes:** Line 213-216, 'We fed the statistical characteristics of the 115 variables into our models, and the simulated carbon emission and removal results were obtained through 10,000 iteration Monte Carlo simulation. Subsequently, statistical analysis was then performed to derive the median and the corresponding lower and upper bounds of the 95% confidence intervals (CI) for the carbon uptake and emission of lime materials.'

**23.Comments:** Ln. 233. Is this value for global scale or for a specific region?

**Response:** The value is for global scale

**Changes:** Line 264-266, 'According to the lime carbon sequestration model, the global uptake of $CO_2$ by lime-containing materials increased from 9.16 Mt C (95% CI:1.84-18.76 Mt C) in 1930 to 34.84 Mt C (95% CI:23.50–49.81 Mt C) in 2020, representing an average annual growth rate of 1.50% (Fig 3a).'

**24.Comments:** Ln. 273-275. Specific values in Gt/yr should be mentioned here.

**Response:** Thank you for your suggestion.

**Changes:** Line 329-333, 'Regarding the global carbon cycle, the annual carbon uptake by lime was approximately 1.65% of the average global forest ecosystem sink from 2001 to 2010, and this can explain approximately 1.55% of the missing global carbon sink (2.37 Gt C yr$^{-1}$) (Data supplement to the Global Carbon Budget 2021, 2022).'

**25.Comments:** Ln. 288. "inconsistent", what do you mean by these results are inconsistent

with those from cement carbon sink? Maybe inconsistent should be changed with another word: "these results contrast with those obtained for the cement carbon sink […]"?

**Response:** Thank you for your modification. Your statement is more scientific and rigorous

**Changes:** Line 348, 'These results contrast with those obtained for the cement carbon sink, where most of the carbon absorption is linked to previous years.'

**26.Comments:** Ln. 294. CO2 -> $CO_2$

**Response:** Chang 'CO2' to '$CO_2$'

**Changes:** Line 355, 'All the original datasets of $CO_2$ uptake by lime are available at https://doi.org/10.5281/zenodo.7628614 (Ma et al.,2023).'

**27.Comments:** Ln. 301-302. Global $CO_2$ emissions reported here in $MtCO_2.yr^{-1}$ are smaller than reported for China in section 3.1. Please, revise.

**Response:** Thank you for your reminding. We have corrected the mistakes

**Changes:** Line 365, 'Global $CO_2$ uptake from lime production processes increased from 9.16 Mt C yr$^{-1}$ in 1930 to 35.27 Mt C yr$^{-1}$ in 2020.'

**CC1**

**General comments:** It seems to me that the use of a principal-component approach is unnecessary and a standard multilinear regression would be simpler, more appropriate, easier to understand by the readers, and also less open to human error. The reason I comment on this is because the results appear to me not make sense. The earliest data point you have for lime production in China is 2002. You estimate the growth of lime production between 1963 and 2002 at a very low 30%, despite growth rates of over 500% for calcium carbide, over 2000% for steel, 800% for completed floor areas, and over 7000% for alumina, the four variables you use in your regression model. It is very difficult to see how lime production could only grow by 30% when all four explanatory variables grow by substantially more than this. Is this reasonable? It implies that even if the uses of lime are extremely low, the production of lime will still be very high, which is illogical. If the growth rate of lime production is greatly underestimated, then the amount of lime produced in the earlier part of the time series here is greatly overestimated, leading to an overestimation of the carbonation uptake. I encourage the authors to look into this again. I would point also to the report by Andrew and Peters on the Global Carbon Project's fossil CO2 dataset, where we try to estimate historical lime production in China. You might disagree with the results, but consider the arguments used anyway. Link to the PDF of this report: https://zenodo.org/record/7215364.

**Response:** The global lime production data came from Lime Statistics and Information (USGS, 2022), but the data did not include the statistics of China's lime production between 1963 and 1984. In addition, the statistical value of China's lime production from 1985 to 2001 was underestimated compared with the actual value. In China, the only source of lime production statistics is the "China Building Materials Industry Yearbook", whose values are provided by the China Lime Association. Its dataset is shown in Figure 1 for actual production, covering the years 1996-2014 and 2019-2020. Considering that the downstream sectors of lime in China mainly include construction, steel, calcium carbide, and alumina, we collected China's cement production (1930-2020), total housing area completed (1963-2020), steel production (1949-2020), calcium carbide production (1949-2020), and alumina production (1954-2020) as variables to fit the lime data. We have adopted the multivariate linear regression method you suggested. In order to avoid multicollinearity, a stepwise

method is used to establish a model. The dependent variable in the model is lime yield, and the independent variable is calcium carbide yield, the completed area of houses in the whole society, cement yield and alumina yield. The determination coefficient of the model is 0.9954, and the adjusted determination coefficient is 0.9942. Given that data on the completed area of housing in the whole society is limited to after 1963, the model only predicts lime production data from 1963-1995 and 2015-2018 (that is, the red line in Figure 1, the shaded part is the 95% confidence interval of the estimation result).

For China's lime production data before 1963, we use the ARIMA model to make predictions. The method is divided into two parts. For 1949-1962, taking China's lime production from 1962 to 2020 as the verification sequence, using the ARIMA (0,1,0) model, taking steel production, calcium carbide production, and cement production as external control variables, the model has a coefficient of determination of 0.9828 (as shown in Figure 2). For the period from 1930 to 1948, using China's lime production data from 1949 to 2020 as the validation sequence, the ARIMA (2,2,0) model without external control variables was used for fitting, and the determination coefficient of the model was 0.9786 (as shown in Figure 3).

[Figure]

Fig 1. China's lime production estimation in1963-2020

[Figure]

Fig 2. China's lime production estimation in 1949-1962

[Figure]

Fig 3. China's lime production estimation in1930-1948

Regarding the different growth rates of lime production and other variables mentioned by reviewer: first, the independent variable we selected is the downstream sector of lime, and there is a significant correlation between these variables and lime production (as shown in Table 1).

**Table 1** Correlation coefficient matrix

|  | Calcium carbide production | Crude steel production | Floor Space Completed | Cement production | Alumina production | Lime production |
|---|---|---|---|---|---|---|
| Calcium | 1 |  |  |  |  |  |

| carbide production | | | | | | |
|---|---|---|---|---|---|---|
| Crude steel production | 0.980** | 1 | | | | |
| Floor Space Completed | 0.951** | 0.974** | 1 | | | |
| Cement production | 0.972** | 0.990** | 0.979** | 1 | | |
| Alumina production | 0.980** | 1.000** | 0.974** | 0.990** | 1 | |
| Lime production | 0.982** | 0.979** | 0.970** | 0.969** | 0.979** | 1 |

**. Correlation is significant at the 0.01 level (2-tailed).

However, although the correlation between the various variables is strong, their growth rates are quite different. Considering the comparability between the various variables, we normalized the time series data of each variable from 1963 to 2020. The normalized growth rate of each variable is shown in Table 2. It can be seen that our fitted lime production growth rate was 341% from 1963 to 2020 and 84% from 1963 to 2002, which was similar to the growth rate of each explanatory variable.

Table 2. Growth rate of each variable after standardization

| | Lime production | Calcium carbide production | Crude steel production | Floor Space Completed | Cement production | Alumina production |
|---|---|---|---|---|---|---|
| Growth rate from 1963 to 2020 | 341% | 416% | 446% | 221% | 338% | 557% |
| Growth rate from 1963 to 2002 | 84% | 60% | 74% | 127% | 101% | 40% |

In the process of fitting, we drew on data provided in the Global Carbon Project's fossil CO2 data. The source of lime yield data in our study is the same as that of Shan et al. (2002-2012) and Cui (2003-2016), namely Almanac of the Chinese Building Materials Industry. In Liu and Wang's study, the lime yield data from 1980 to 1990 was mentioned, but as mentioned in your discussion, this part of the data is being underestimated, and we also used our method to fit according to this data as a verification sequence, but the result was not what we hoped, and the yield of the fitted value even appeared negative. Regarding Olivier's research, I'm sorry we didn't find the reference, but through your report, we learned that the research interpolates the lime production in China. The first key reference point is China's first national communication in 1994. Since the national communication has only two years of data, and Shan et al. pointed out in their study that the data on lime production in these two reports were unreasonable and regarded as abnormal values, they were not used in our study. It was used as a reference.

It is worth mentioning that, in our estimation, China's lime production peaked in 1986, and showed

a downward trend from 1986 to 1990. During this period, China's "Seventh Five-Year Plan" period, the government carried out timely rectification of the overheated economic development in the previous period. It can also be seen from the data of various variables that during this period, the area of completed housing in the whole society has dropped significantly, the output of calcium carbide has fluctuated, and the output of cement and alumina has changed. The trend is similar with study by Liu and Wang et al.

[revised manuscript text omitted]

---

## Referee Report (RR1)

It was very difficult to follow your manuscript corrections and the changes related to reviewers' comments. The authors answered some questions and did some major corrections, but there are still several points that need to be developed in the manuscript. Additionally, it does not seem correct to say and conclude that lime should be considered as a carbon sink when the results show lime as a carbon source, as already mentioned in the previous comments. Please find my comments below.

**General and technical comments**

Ln. 12. Needs to be rewritten. Suggestion: "However, during the life cycle of lime production, the alkaline components of lime will continuously absorb $CO_2$ from the atmosphere during use and waste disposal."

It seems important to add standard deviation for all your result values. You mention the annual carbon sequestration of lime accounts for about 1.03±?? %, what would be the standard deviation associated to this estimate for example?

Ln.330. The annual carbon uptake by lime represents 1.65% of the global forest sink. This will result in a very low global annual carbon sink then. The global carbon sink is the sum of carbon sink from ocean, land and cement. So, the 1.65% of the global forest sink cannot be equal to 1.65% of the global carbon sink. Consequently, lime sink cannot explain the 1.55% of the missing global carbon sink. This paragraph does not seem correct. The net emissions of lime production and impact on global carbon budget is an important point that needs to be developed in your study. Please develop.

You mentioned in your reply that you used auto-regressive models and other methods to predict the data from 1963 to 2000, but the method used in your study obtained the largest coefficient of determination. This seems important enough to be mentioned in your manuscript. I would suggest adding the different methods coefficient in supplement information to justify why you used the linear regression method for your study.

Supplement information SI data 4 of "Lime material production and uses", how are calculated current year, previous year and total? What do the values represent? Captions to the different tables are missing.

It would be useful to precise which supplementary information table need to be read in the manuscript instead of just mentioning "see the Supplementary Information". Additionally, might be better to have the supplementary Information in PDF as well.

Figure 1. What are the meaning for the different colors? You mention in Figure caption "double solid lines", I do not see any.

Ln. 257. "This figure is higher" do you mean the results of the figure are higher?
How can your results be higher than Tong et al., 2019 if you considered an emission reduction scenarios?

The carbon sink increases with time but because the production has increase. This increase for both the sink and the emission seems to be proportional to each other. This should be mentioned in the discussion.

Ln. 360. It is not correct to say that lime should be considered as a carbon sink. The net emissions show a carbon source.
One of your conclusion should be that the sink associated with lime life cycle should not be neglected and should be considered for future carbon cycle studies. However, there are still some questions not answered in your study about the emission inventories used here which could overestimate or underestimate lime production or lime sink and make your results biases. More development should appear in your discussion regarding these aspects.

---

## Author Response (AR2)

Thank you for your precious comments and suggestions. Those comments are all valuable and very helpful for revising and improving our paper, as well as the important guiding significance to our researches. The responds to the reviewer's comments are as following:

1. **Comments**: Ln. 12. Needs to be rewritten. Suggestion: "However, during the life cycle of lime production, the alkaline components of lime will continuously absorb $CO_2$ from the atmosphere during use and waste disposal."

**Response:** Thank you for your valuable suggestions on how to improve this sentence.

**Changes:** The sentence in line 12 has been rewritten.

2. **Comments**: It seems important to add standard deviation for all your result values. You mention the annual carbon sequestration of lime accounts for about 1.03±?? %, what would be the standard deviation associated to this estimate for example?

**Response:** The lime carbon sequestration model uses 115 input parameters, which follow statistical distributions such as normal, uniform, and triangular distributions. Due to the non-linear simulation process, the overall distribution is unclear. To provide more information about parameter estimation properties (Efron, 1982), we have utilized the percentile CI method to estimate the uncertainty range. This method determines the upper and lower limits of the interval based on the 2.5th and 97.5th percentiles. For further information, please refer to the following reference: https://epubs.siam.org/doi/book/10.1137/1.9781611970319.

**Changes:** None.

3. **Comments**: Ln.330. The annual carbon uptake by lime represents 1.65% of the global forest sink. This will result in a very low global annual carbon sink then. The global carbon sink is the sum of carbon sink from ocean, land and cement. So, the 1.65% of the global forest sink cannot be equal to 1.65% of the global carbon sink. Consequently, lime sink cannot explain the 1.55% of the missing global carbon sink. This paragraph does not seem correct. The net emissions of lime production and impact on global carbon budget is an important point that needs to be developed in your study. Please develop.

**Response:** Thank you for your valuable suggestion. We appreciate your insight and have carefully considered it. Upon further review, we have decided to revise the section you mentioned. We now recognize that quantitative comparisons between different types of carbon sinks may have little practical significance. Therefore, we have chosen to remove this part from the updated version of our content.

**Changes:** Ln 349-352, 'Regarding the global carbon cycle, lime's annual carbon uptake is estimated to be approximately 1.09% of the average global land carbon sink from 2010 to 2020, which was approximately 3.18 Gt C yr-1 (Global Carbon Budget, 2022). This indicates that lime's contribution to the global carbon cycle is significant and should be taken into account when considering strategies to mitigate carbon emissions.'

4. **Comments**: You mentioned in your reply that you used auto-regressive models and other methods to predict the data from 1963 to 2000, but the method used in your study obtained the largest coefficient of determination. This seems important enough to be mentioned in your manuscript. I would suggest adding the different methods coefficient in supplement information to justify why you used the linear regression method for your study.

**Response:** Thank you for your valuable feedback, which we have incorporated into our manuscript. We have added the parameters statistics of various regression models to the following tables, as per your suggestion, and marked the changes in line 90.

**Changes:** See the following tables for details (SI-2 Data 4)

Statistics of estimated parameters in regression models for predicting lime production in China 1963-1995.

| Model | Regression method | Dependent Variable | Independent variable | | Estimate Std. | Error | t value | Sig. | F-statistic | p-value | R-square | Adjusted R-square |
|---|---|---|---|---|---|---|---|---|---|---|---|---|
| 1 | Principal Component Regression | Lime production of China from China Construction Material Industry Yearbook | Production of calcium carbide, crude steel, cement and alumina, and completed area | (Intercept) | 142.093 | 3.021 | 47.03 | < 2e-16*** | 306.9 | 5.20E-14 | 0.9360 | 0.9329 |
| | | | | Z1[1] | 22.582 | 1.289 | 17.52 | 5.2e-14*** | | | | |
| 2 | | Lime production of China from USGS | Production of calcium carbide, crude steel, cement and alumina, and completed area | (Intercept) | 75.516 | 6.118 | 12.34 | 4.55e-13*** | 378.9 | <2.2e-16 | 0.9289 | 0.9265 |
| | | | | Z1 | 45.674 | 2.346 | 19.46 | <2e-16*** | | | | |
| 3[2] | Stepwise Linear | Lime productio | Production of calcium | (Intercept) | 81.96 | 8.19 | 10.008 | 2.72e-08*** | 862.6 | <2.2e-16 | 0.9954 | 0.9942 |

| | Regression | n of China from China Construction Material Industry Yearbook | carbide, cement and alumina, and completed area | calcium carbide | 4.096 | 0.67 | 6.113 | 1.50e-05*** | | | | |
| | | | | completed area | 0.035 | 0.007 | 5.232 | 8.22e-05*** | | | | |
| | | | | cement | -0.054 | 0.01 | -5.607 | 3.94e-05*** | | | | |
| | | | | alumina | 0.013 | 0.001 | 9.254 | 7.99e-08*** | | | | |
| 4 | | Lime production of China from USGS | Production of crude steel | (Intercept) | 4.53887 | 7.72733 | 0.587 | 0.561 | 472.7 | <2.2e-16 | 0.9422 | 0.9402 |
| | | | | calcium carbide | 0.30844 | 0.01419 | 21.741 | <2e-16*** | | | | |

1, One principal component (Z1) is obtained in principal component analysis

2, Model 3 was selected as the best regression method in this study

*** indicates a significant difference (p<0.001)

Fitted coefficients and standard errors of ARIMA models for predicting lime production in China 1949-1962.

| Model | Fitting method | external regressor variable | | Coefficients | Standard error | R-square |
|---|---|---|---|---|---|---|
| 5 | ARIMA(0,1,0) | Production of calcium carbide, cement and crude steel | crude steel | 0.1368 | 0.0467 | 0.9828 |
| | | | cement | -0.0588 | 0.0163 | |
| | | | calcium carbide | 3.7173 | 0.9585 | |

Fitted coefficients and standard errors of ARIMA models for predicting lime production in China 1930-1948

| Model | Fitting method | External regressor variable | | Coefficients | Standard error | R-square |
|---|---|---|---|---|---|---|
| 6 | ARIMA(0,2,2) | Without | MA1 | -1.1748 | 0.1497 | 0.9786 |
| | | | MA2 | 0.2902 | 0.1604 | |

Fitted coefficients and standard errors of ARIMA models for predicting lime production in global 1930-1962

| Model | Fitting method | External regressor variable | | Coefficients | Standard error | R-square |
|---|---|---|---|---|---|---|
| 7 | ARIMA(1,0,0) | Production of global crude steel, cement, and alumina | AR1 | 0.7929 | 0.0787 | 0.9849 |
| | | | intercept | 131322.7 | 11798.17 | |
| | | | crude steel | 0.0871 | 0.0274 | |
| | | | cement | -0.0082 | 0.0104 | |
| | | | alumina | 1.0343 | 0.4059 | |

5. **Comments**: Supplement information SI data 4 of "Lime material production and uses", how are calculated current year, previous year and total? What do the values represent? Captions to the different tables are missing.

**Response:** Thank you for your suggestion. We have readjusted the table and added corresponding notes. See SI-1 Data 4 of "Lime carbon emission and uptake results" for details. (https://doi.org/10.5281/zenodo.7759053)

**Changes:** See SI-1 Data 4 of "Lime carbon emission and uptake results" for details.

'1. From a vertical perspective, the sum of the vertical data in the table represents the annual carbon sequestration of lime-based materials, which is the total amount of carbon sequestered. The diagonal data indicates the carbon sequestration amount of lime-based materials produced in the current year, whereas the amount of carbon sequestration in previous years can be calculated by subtracting the current year's value from the annual carbon sequestration.(For example, in 1935, the annual carbon sequestration of lime-based materials was 8.11651 million tons (Mt), of which 7.83204 Mt was due to the carbonization of lime-based materials produced in the same year. The remaining carbon sequestration amount of lime-based materials produced in the years 1930-1934 were 0.00149, 0.00135, 0.00116, 0.00245, and 0.34222 Mt, respectively, adding up to a total of 0.33448 Mt in historical years.);

2.Horizontally, the horizontal data refers to the annual carbon sequestration of lime-based materials produced in a certain year over time. Taking 1935 as an example, in addition to the carbonation that occurred in that year, the lime-based materials produced in 1935 absorbed $CO_2$ from the atmosphere annually from 1936 to 2020, with the amount of $CO_2$ absorbed declining from 0.31732Mt in 1936 to 0.00023Mt in 2020.'

6. **Comments**: It would be useful to precise which supplementary information table need to be read in the manuscript instead of just mentioning "see the Supplementary Information". Additionally, might be better to have the supplementary Information in PDF as well.

**Response:** Thank you for your suggestion. We have incorporated your feedback into our manuscript. Specifically, we have added statements about specific tables to the supplementary information as per your suggestion. Additionally, we have also included an introduction to the attachment information in the "Data availability" section. We appreciate your input and guidance, and thank you for taking the time to review our work."

**Changes:** Ln 380-401

'SI-1 Lime carbon emission and uptake results, 1930-2020

Data 1. Annual carbon uptake by lime material and region

Data 2. Global carbon uptake by lime material and stage

Data 3. Global carbon uptake by region

Data 4. Annual global carbon uptake by lime material and relevant lag time, 1930 to 2020

Data 5. Cumulative process $CO_2$ emissions from lime production by region and category, 1930 to 2020

Data 6. Global process $CO_2$ emissions from lime production and carbon uptake by lime materials carbonation from 1930 to 2020

SI-2 Lime material production and uses, 1930-2020

Data 1. Lime production by region, 1930 to 2020

Data 2. Estimated production of lime in China, 1930 to 2020

Data 3. Estimated global lime production, 1930 to 2020

Data 4. Parameters of lime production fitting model

Data 5. Paper and paperboard production by region, 1930 to 2020

Data 6. Steel production by region, 1930 to 2020

Data 7. Alumina production by region, 1930 to 2020

Data 8. Output rate by material

Data 9. Estimates of lime used for different industries by region

SI-3 Uncertainty of lime carbon emission and uptake, 1930-2020

Data 1. Variables considered in the uptake uncertainty analysis using a Monte Carlo method

Data 2. The uncertainty of $CO_2$ emissions from lime production

Data 3. The uncertainty of lime carbon uptake'

**7. Comments**: Figure 1. What are the meaning for the different colors? You mention in Figure caption "double solid lines", I do not see any.

**Response:** The figure uses different colors to indicate various lime-based materials that are capable of absorbing $CO_2$ during different life cycles. Descriptions of different colors are supplemented in the notes to Figure 1. Additionally, the outermost border in the figure is indicated by 'double solid lines'

**Changes:** we have made modifications to Figure 1 and its annotations (Ln 122-127). Specifically, the changes we made are as follows:

[Figure]

**Figure 1: System boundary for the sequestration of carbon by lime. Solid arrows represent the material flow, dashed arrows indicate the carbon flow. (Yellow, blue, and red represent lime-based materials with carbon absorption capacity and their associated production processes, spanning from initial production through usage and waste disposal. Gray represents materials, production processes, or disposal methods with little carbon absorption capacity. Green represents the disposal method for lime-based waste that possesses carbon absorption potential.)**

**8. Comments**: Ln. 257. "This figure is higher" do you mean the results of the figure are higher? How can your results be higher than Tong et al., 2019 if you considered an emission reduction scenario?

**Response:** Thanks very much for the opinion. Ln. 257 of the article compares our calculated data

on lime carbon emissions in 2020 (49.93Mt C yr$^{-1}$) with the Tong et al., 2019, in which China's lime carbon emissions in 2020 are predicted to be 46.91Mt C yr$^{-1}$, but this forecast considered an emission reduction scenario, they assumed the technology penetration rate of CCU would reach 5% by 2020 in China. However, as of 2020, the CCU technology was rarely utilized in China's lime industry. Consequently, the amount of carbon emissions produced by lime manufacturing is likely higher than in a scenario where CCU technology is used. Therefore, it is reasonable to assume that our calculations may have overestimated the carbon emissions generated by this industry. Our statement may not be clear enough, and we have revised it in Ln. 259.

**Changes:** Ln 260-264, 'The current figure exceeds the 46.91 Mt C yr-1 forecasted for 2020 by Tong et al. (2019), which can be attributed primarily to the emission reduction scenarios they considered, assuming a technology penetration rate of 5% for CCU in China by 2020. However, it is important to note that as of 2020, CCU technology was seldom employed in China's lime industry. Therefore, the actual amount of carbon emissions produced by lime manufacturing is likely to be higher than in the scenario considered by Tong et al. Thus, our calculations are reasonable.'

9. **Comments**: The carbon sink increases with time but because the production has increase. This increase for both the sink and the emission seems to be proportional to each other. This should be mentioned in the discussion.

**Response:** Thank you for your suggestion. We have added a description of this section to the discussion of the new revised manuscript.

**Changes:** Ln 335-336, 'The carbon sink increases over time, but this increase is due to an increase in production. It seems that both the increase in the sink and the emissions are proportional to each other.'

10. **Comments**: Ln. 360. It is not correct to say that lime should be considered as a carbon sink. The net emissions show a carbon source.

**Response:** Thank you for your opinion. We have revised this expression to make it more scientific

**Changes:** Ln 404-405, 'The national greenhouse gas inventories guideline and global carbon budgets could be improved by accounting for lime uptake, which can offset approximately 38% of emissions from industrial lime processes.'

11. **Comments**: One of your conclusions should be that the sink associated with lime life cycle should not be neglected and should be considered for future carbon cycle studies. However, there are still some questions not answered in your study about the emission inventories used here which could overestimate or underestimate lime production or lime sink and make your results biases. More development should appear in your discussion regarding these aspects.

**Response:** Thank you very much for your suggestion. It will be extremely valuable in guiding the revision of our paper.

**Changes:** Ln 335-348, 'The carbon sink increases over time, but this increase is due to an increase in production. It seems that both the increase in the sink and the emissions are proportional to each other. Our research results on carbon emissions and carbon absorption are significantly impacted by lime production. However, due to the lack of available data on annual lime production in China and worldwide during the early years, we used fitting methods to fill the gap of lime production and estimate it up to 1930. The statistically inferred 95% confidence interval was then used as the uncertainty range for lime production. To incorporate this uncertainty range into the accounting model for carbon sequestration and carbon emissions, we

used Monte Carlo simulations, and after 10,000 iterations, we obtained the final accounting results for carbon sequestration and carbon emissions. Therefore, from the interpolation of production data to the final accounting of carbon sinks and carbon emissions, all potential sources of uncertainty have been fully considered in the accounting process. Thus, this is a crucial way to obtain lime carbon sink and carbon emissions data from 1930 to 2020 under current data conditions. However, as our understanding of basic data and the mechanisms of lime production, carbon sequestration, and carbon emissions deepens, and as we improve our activity level data, such as lime-based material utilization, waste stacking, and recycling rates, and optimize carbonization parameters under different exposure conditions, there is still considerable potential for improving the accuracy of long time series lime material carbon sequestration and carbon emission accounting.'

---

## Author Response (AR3)

Thank you for your valuable feedback on our manuscript. We appreciate your comment and have taken your suggestion into consideration. Specifically, we have made the recommended changes to the paper as per your suggestion. We believe that this will improve the manuscript and thank you for helping us to enhance the quality of our work. Once again, we thank you for your time and effort in reviewing our manuscript.

**1. Comments:** Ln. 351. "This indicates that lime's contribution to the global carbon cycle is significant and should be take into account when considering strategies to mitigate carbon emissions." I would suggest writing this sentence as "This indicates that lime production contributes to the global carbon cycle and should be taken into account for better accuracy in national emission inventories."

**Response:** Thank you for your valuable feedback on our previous draft. We appreciate your suggestion and have made the necessary correction accordingly.

**Changes:** The sentence in line 351 has been rewritten. "This indicates that  $CO_2$  uptake by lime contributes to the global carbon cycle and should be taken into account for better accuracy in national emission inventories."